**Multiple thermal AMOC thresholds in the intermediate complexity model Bern3D**

Markus Adloff[1,2*], Frerk Pöppelmeier[1,2], Aurich Jeltsch-Thömmes[1,2], Thomas F. Stocker[1,2], Fortunat Joos[1,2]

[1] Centre for Environmental Physics, University of Bern, Switzerland

[2] Oeschger Centre for Climate Change Research, University of Bern, Switzerland

*Contact: markus.adloff@unibe.ch

**Abstract**

Variations of the Atlantic Meridional Overturning Circulation (AMOC) are associated with Northern Hemispheric and global climate shifts. Thermal thresholds of the AMOC have been found in a hierarchy of numerical circulation models, and there is an increasing body of evidence for the existence of highly sensitive AMOC modes where small perturbations can cause disproportionately large circulation and hence climatic changes. We discovered such thresholds in simulations with the intermediate complexity Earth system model Bern3D, which is highly computationally efficient allowing for studying this non-linear behaviour systematically over entire glacial cycles. By simulating the AMOC under different magnitudes of orbitally-paced changes in radiative forcing over the last 788,000 years, we show that up to three thermal thresholds are crossed during glacial cycles in Bern3D, and that thermal forcing could have destabilised the AMOC repeatedly. We present the circulation and sea ice patterns that characterise the stable circulation modes between which this model oscillates during a glacial cycle, and assess how often and when thermal forcing could have preconditioned the Bern3D AMOC for abrupt shifts over the last 788 kyr.

## 1 Introduction

The Atlantic Meridional Overturning Circulation (AMOC) transports warm waters from the Southern Hemisphere and the Mexican Gulf towards the Nordic Seas, until the gradually cooled salty water lost enough buoyancy and sinks, forming North Atlantic Deep Water (NADW). This water mass moves southwards along the western boundary of the Atlantic until it encounters the denser Antarctic Bottom Water (AABW) and slowly rises and upwells in the Southern Ocean, being ultimately incorporated either into AABW or the lighter Antarctic Intermediate Water (AAIW). The northward heat transport of the AMOC shapes regional climate by pushing the polar front north by several degrees of latitude, effectively

producing a climate in Europe and Greenland that is milder than predicted from latitude/insolation alone (Ruddiman and McIntyre 1981, Bard et al. 1987). It also affects global climate by shifting the Intertropical Convergence Zone (ITCZ) and monsoon systems (Wang et al., 2001, Bozbiyik et al, 2011), and interacting with the regional climate and deep water formation in the North Pacific (Okazaki et al., 2010, Menviel et al., 2012, Praetorius and Mix, 2014). The AMOC furthermore shapes biological surface productivity by regulating nutrient supply to the surface ocean in the Atlantic and Pacific (Tetard et al., 2017, Joos et al., 2017). On its southward path in the Atlantic, it influences deep ocean nutrient, carbon, and oxygen concentrations (Broecker, 1991). By affecting primary production and deep ocean carbon storage, AMOC changes also modulate atmospheric greenhouse gas concentrations (e.g. Menviel et al., 2008). Rapid changes in AMOC and hence Atlantic heat and carbon redistribution occurred repeatedly during the last glacial, termed Heinrich (Heinrich, 1988, Broecker, 1994) and Dansgaard-Oeschger events (Oeschger et al., 1984, Dansgaard et al., 1993), which had regional and global impacts on ecosystems and humans (e.g. Severinghaus et al., 2009, Timmermann and Friedrich, 2016). Yet, the factors determining AMOC stability are not fully understood.

As part of the thermohaline circulation, the AMOC is sensitive to both salinity and thermal forcing. Depending on the location of deep water formation in both hemispheres, the AMOC can switch between stable circulation states - either gradually or abruptly - as local vertical density profiles, sea ice extent, and meridional heat and salinity gradients change. Numerical experiments showed that large freshwater inputs into the North Atlantic can theoretically cause abrupt shifts from a vigorous circulation state to a temporarily subdued or collapsed circulation (e.g. Stocker and Wright, 1991, reviews by Weijer et al., 2019, Jackson et al., 2023). Such possible shifts of circulation state were first identified in box models (Stommel 1961) and confirmed in intermediate complexity models and global circulation models (Jackson and Wood, 2018, review in Jackson et al, 2023). AMOC bistability could explain reconstructed sudden AMOC state shifts in the Pleistocene, possibly caused by large freshwater fluxes from melting continental ice shields and increased iceberg transport into the North Atlantic at the onset of Heinrich Events (Broecker, 1994, Grousset et al., 2000). Lags between the appearance of ice-rafted debris and the reconstructed cooling, however, suggest that freshwater fluxes could have instead acted as a positive feedback to AMOC weakening rather than triggering it (Barker et al., 2015).

Besides Heinrich event-like AMOC shifts to a less vigorous circulation in response to strong freshwater forcing, there is increasing evidence for metastable AMOC states in-between the glacial and interglacial circulation end-members. In some numerical models, and for narrow

parameter ranges (e.g. atmospheric $CO_2$ concentrations, ice sheet configurations), the AMOC in such intermediate climate states is sensitive to small internal or external variability (e.g. Aeberhardt et al., 2000, Knutti et al., 2002, Zhang et al., 2014b, Zhang et al., 2017) and can sustain spontaneous oscillations ( Brown and Galbraith, 2016, Vettoretti et al., 2022, Armstrong et al., 2022, review of CMIP6 models in Malmierca-Vallet et al., 2023). Some of these oscillations could be analogues to Dansgaard-Oeschger events that have been identified during intermediate glacial climate conditions, specifically during Marine Isotope Stage (MIS) 3, and are thought to be caused by internal feedbacks that amplified small changes of the North Atlantic salinity balance (Zhang et al., 2014, Zhang et al., 2014b, Zhang et al., 2017, Klockmann et al., 2020, Vettoretti et al., 2022, Armstrong et al., 2022). Meteoric and terrestrial freshwater input to the surface ocean are climate-dependent, as is ice rafting and the salt rejection associated with sea ice formation. These processes are thus impacted by, and impact themselves, the AMOC (Ganopolski and Rahmstorf, 2001, Barker et al., 2015). Feedbacks similarly exist for the salinity transport from the tropics to the North Atlantic, global circulation patterns, and the salinity gradients which determine salt transport into the Atlantic basin through the Bering Strait, Drake Passage, and from the Indian Ocean (e.g. Rahmstorf 1996). Besides salinity changes, numerical experiments with GCMs also showed that the vertical temperature profile affects AMOC stability (Haskins et al., 2020). Short-term AMOC weakening in response to warming has been simulated by a wide range of GCMs (e.g. Mikolajewicz et al., 1990, Gregory et al., 2005, Weijer et al., 2020). Thermal forcing of the North Atlantic has also been found to cause longer term gradual changes in AMOC strength in intermediate and higher resolution models (Manabe and Stouffer, 1993, Stocker and Schmittner, 1997, Knorr and Lohmann, 2007, Zhang et al., 2017, Galbraith and Lavergne, 2019). In addition, bistability of AMOC under thermal forcing has been found in uncoupled and coupled GCMs (Oka et al., 2012, Klockmann et al., 2018), and thermal forcing, especially of the Southern Ocean, can cause abrupt AMOC state transitions similar to hosing in the North Atlantic (Oka et al., 2021, Sherriff-Tadano et al., 2023). An important process in the cooling-driven weakening of AMOC is the covering of former deep convection sites with sea ice, which then causes a southward shift of deep convection (Oka et al., 2012). Such a southward shift is only possible if the water column south of existing convection sites is sufficiently destabilised by climate-driven density changes (Ganopolski and Rahmstorf, 2001).

So far, simulations of thermal AMOC thresholds have mostly been conducted with computationally expensive numerical models, and the implications of the existence of AMOC instability and thermal thresholds have not been tested across entire glacial cycles. While providing crucial process understanding, the limited simulation length makes direct

comparisons of these simulations to proxy timeseries challenging, which is required to assess the role of these processes in glacial-interglacial AMOC changes. The existence of multiple AMOC equilibria seems to be determined by the model-dependent existence and strength of feedbacks, with more complex models including more, possibly counteracting, feedbacks (Weijer et al., 2019). Yet, systematic testing of AMOC stability and long transient simulations are done more easily in lower complexity models than General Circulation Models (GCMs).

Here, we demonstrate the existence of hysteresis and mode shifts in the AMOC in the computationally-efficient, intermediate complexity model Bern3D under radiative forcing. The model can be used to study AMOC changes with and without freshwater hosing over full glacial cycles. We provide a comprehensive description of the underlying processes of the simulated AMOC response to radiative changes and elucidate their influence on the AMOC dynamics during orbitally-forced glacial-interglacial cycles in transient simulations of the last eight glacial cycles.

## 2  Methods

We employed the Bern3D intermediate complexity model version 2.0 (Müller et al., 2006, Roth et al., 2014) to investigate the AMOC behaviour under a wide range of radiative forcing. The Bern3D model comprises a 3D ocean component with a 40x41 horizontal grid and 32 depth layers, along with a 2D atmosphere (spatially-explicit energy-moisture balance with prescribed wind fields) and dynamic sea-ice. The model explicitly calculates the thermo-haline circulation with a frictional-geostrophic flow (Edwards et al., 1998) and contains parameterizations to account for isopycnal diffusion and eddy-turbulence via the Gent-McWilliams parameterization (Griffies, 1998). Temperature and salinity are dynamically transported by the physical ocean model and respond to static seasonal wind fields and changing atmospheric 2D energy and moisture balance, sea ice formation and external forcings. Bern3D explicitly calculates Pacific-Atlantic transport through the Bering Strait, and freshwater flux corrections are only imposed in the Weddell Sea, and compensated for in the Southern Ocean to induce stronger deep water formation (Ritz et al., 2011, Roth et al., 2014).

Table 1: Overview of the model experiments in this study. In set A, radiative forcing from dust is scaled linearly with $\delta^{18}O$ and assuming different magnitudes at LGM as given in parentheses.

| Simulation Set | Simulation ID | Starting point and length | Forcing | Purpose |
|---|---|---|---|---|
| A | A0 | MIS 19 spin-up 787500 years | orbital+GHG+dust(0 W/m$^2$) | test AMOC changes in response to transient glacial-interglacial radiative forcing |
| | A1 | | orbital+GHG+dust(-1 W/m$^2$) | |
| | A2 | | orbital+GHG+dust(-2 W/m$^2$) | |
| | A3 | | orbital+GHG+dust(-3 W/m$^2$) | |
| | A4 | | orbital+GHG+dust(-4 W/m$^2$) | |
| | A5 | | orbital+GHG+dust(-5 W/m$^2$) | |
| | A6 | | orbital+GHG+dust(-6 W/m$^2$) | |
| | A7 | | orbital+GHG+dust(-7 W/m$^2$) | |
| | A8 | | orbital+GHG+dust(-8 W/m$^2$) | |
| B | B.slow | PI spin-up, 105 kyr | linear change in RF from 0 to -10 W/m$^2$ over 50 kyr and recovery over next 50 kyr | identify processes that cause AMOC shifts under radiative forcing |
| | B.slow.a | year 23000 of B.slow, 20 kyr | 0.1 Sv freshwater input over 100 yr | test AMOC stability at different time steps in B.slow |
| | B.slow.b | year 24500 of B.slow, 20 kyr | | |
| | B.slow.c | year 28500 of B.slow, 5 kyr | | |
| | B.slow.d | year 47000 of B.slow, 5 kyr | | |
| | B.fast.PI | PI spin-up, 25 kyr | linear change in RF from 0 to -10 W/m$^2$ over 10 kyr and recovery over next 10 kyr with different orbital parameters | test dependence of AMOC response to radiative forcing to orbital constellation |
| | B.fast.21ka | PI spin-up, 25 kyr | | |
| | B.fast.30ka | PI spin-up, 25 kyr | | |
| | B.fast.50ka | PI spin-up, 25 kyr | | |
| | B.fast.80ka | PI spin-up with, 25 kyr | | |

We conducted two sets of simulations with the Bern3D model (Table 1). In set A, comprising nine simulations, we fully transiently simulated the last 788 kyr by imposing changes in orbital configuration, ice sheet albedo, and globally-averaged radiative forcing from the well-mixed greenhouse gases (GHG) $CO_2$ and $CH_4$ (combined here labelled as the 'standard forcing'). The runs started from an interglacial steady state (50 kyr with pre-industrial (PI) conditions and 2 kyr of re-adjustment to the radiative balance of MIS 19c). Orbital (Berger, 1978, Berger and Loutre, 1991), GHG (Bereiter et al., 2015, Loulergue et al., 2008, Joos and Spahni, 2008), and ice sheet albedo forcing (i.e. the standard forcing) is identical in each run (Fig. 1). Ice sheet albedo changes are calculated based on the benthic $\delta^{18}O$ LR04 stack (Lisiecki & Raymo, 2005) smoothed by averaging over a 10000-year moving window for the past 788 kyr.

The LR04 stack was chosen because it is the only complete record with constant temporal resolution over the simulated period. In our experiments, we applied spatially-uniform radiative forcings, to account for uncertainties in the glacial radiative balance, e.g. uncertain atmospheric optical depth changes due to changes in aerosols and dust, in addition to the better constrained temperature changes due to orbital changes and greenhouse gases, hence termed dust forcing. The scale of this forcing varies between the simulations and transiently within each simulation. The maximum radiative dust forcing, defined via the peak

LGM value in the smoothed $\delta^{18}O$ stack, is a free parameter, ranging from 0 to -8 W/m$^2$
relative to PI (Simulations A.0 to A.8). To construct the forcing, we scaled the maximum
forcing linearly with the smoothed LR04 stack, given the close correlation of reconstructed
dust fluxes and ice volume likely due to the dominant role of wind fields, sea level, and
hydrological cycle on dust fluxes (Winckler et al., 2008). The range of the resulting combined
radiative forcing is between -3 and -10 W/m$^2$. This range brackets estimates of maximum
reductions in global mean radiative forcing at the LGM of 7 - 8 W/m$^2$ due to albedo,
greenhouse gas, and aerosol effects (Albani et al., 2018). The imposed forcings resulted in
global mean surface temperature (GMST) differences between the LGM and PI of -3 to -9.6
°C. This temperature range encompasses most of the LGM-PI range reported in studies
investigating the Paleo Model Intercomparison Project (PMIP) 2, PMIP3, and PMIP4, which
range from -3.1 to -7.2 °C (Masson-Delmotte et al., 2013, Kageyama et al., 2021).

Furthermore, these simulations are also consistent with proxy-based reconstructions that
indicate GMST differences between -2 and -8 °C (Tierney et al., 2020), as well as covering
the -6.1 °C GMST difference as constrained by a recent data assimilation study with the
CESM model (Tierney et al., 2020). It is important to note that we only considered the
radiative effect of an assumed uniform distribution of aerosols in our simulations. In reality,
this distribution would be non-uniform and aerosols would have additional effects on
atmospheric freshwater fluxes, two factors which are both relevant for AMOC stability
(Menary et al., 2013) but are poorly constrained for the last 788 kyr. Furthermore, freshwater
fluxes associated with the build-up and disintegration of continental ice sheets and glaciers
were not taken into account in any of the simulations presented here. We also kept the
topography constant and do not close the Bering Strait during glacial states.

Simulation set B (Tab. 1) was designed to investigate the mechanisms behind radiation-
driven AMOC changes under more idealised boundary conditions. This simulation set
includes one long run with "slowly" changing radiative forcing to a peak of -10 W/m$^2$ (105 kyr,
B.slow), five short simulations with "fast" changing forcing (25 kyr, B.fast), and four
simulations branched off from B.slow at different points in time. B.slow started from a pre-
industrial state, followed by a linearly decreasing negative radiative forcing over 50 kyr,
followed by a linear increase of forcing back to the initial state also over 50 kyr (Figure 4).
We continued the simulation for an additional 5 kyr under constant, pre-industrial conditions
to let the model re-equilibrate. The magnitude of this forcing is on the uppe end of the range
explored in simulation set A (A6-A8).
The setup of B.fast.PI is analogous to B.slow with the radiation decrease and consecutive
increase spanning 20 kyrs. The simulations started from a steady state with pre-industrial
orbital and GHG configuration, and were run with orbital configurations of PI, 21, 30, 50 and
80 kyrBP (simulations B.fast.PI, B.fast.21ka, B.fast.30ka, B.fast.50ka, B.fast.80ka,
respectively).

At four specific time points in B.slow, we branched off simulations to test the AMOC stability
by keeping all forcings constant, but at the same time applying a small freshwater hosing to
the North Atlantic (45°N-70°N) with a magnitude of 0.1 Sv over 100 years. If the AMOC is in
a stable mode i.e. far from a bifurcation point, it should recover from these freshwater
perturbations returning to its initial strength, while an unstable AMOC close to a bifurcation
point should transition into a new circulation mode.

We incorporated three passive circulation tracers ('dyes') in set B. Each of these dye tracers
is restored to 1 at the surface of a chosen region (Fig. SI.1), and to zero elsewhere in the
surface ocean, and has no sources or sinks below the surface. In the deep ocean, the dye
tracer concentration is hence diluted only by mixing with other water masses sourced from
other regions. These artificial dye tracers allow us to track the dispersal of North Atlantic
Deep Water (NADW), Antarctic Intermediate Water (AAIW) and Antarctic Bottom Water
(AABW) in the ocean interior.


**3   Results and Discussion**

We first investigate the response of the AMOC to changes in orbital configuration and
radiative forcing as transiently simulated in our 788 kyr-long simulations of set A. We aim to
provide a comprehensive understanding of radiation-driven AMOC dynamics on glacial-
interglacial timescales. Subsequently, we utilise the more idealised setup of simulation set B
to further examine the underlying mechanisms driving these changes in more detail.


**3.1. AMOC changes over the past eight glacial cycles**

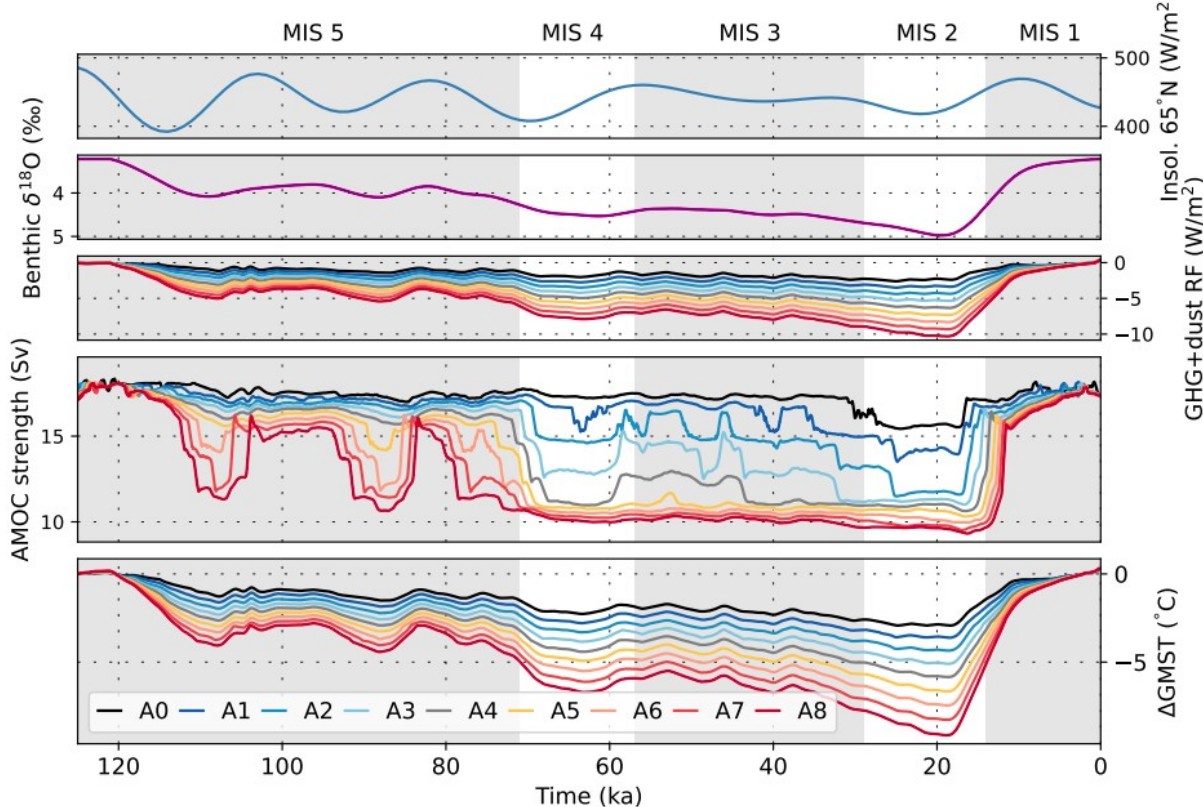

Figure 1: Forcings, AMOC and temperature response over the last 125 kyr of simulation ensemble A. The upper three panels show July Insolation at 65°N, benthic $\delta^{18}$O (10 kyr spline of LR04, Lisiecki and Raymo, 2005) used to scale the dust forcing and the combined effect of our dust forcing for each simulation and reconstructed atmospheric $CO_2$ changes (Bereiter et al., 2015), smoothed with a second-order lowpass filter (cutoff frequency: 1/2000). The lower two panels show the 500 yr running mean of simulated AMOC strength and GMST deviations from the PI in every simulation of simulation set A. Colours in the lower three panels differentiate between simulations with different amplitudes of the radiative forcing (see Methods).

In our simulations, radiative forcing- and orbitally-driven temperature changes resulted in both gradual and abrupt AMOC shifts during each of the last eight glacial cycles (Fig. SI.2). Fig. 1 illustrates the simulated AMOC threshold behaviour during these changes over the entire last glacial cycle (past 125 kyr) with the different dust forcing scalings. Abrupt changes in AMOC strength occurred in every simulation, with larger changes occurring under stronger forcing. The magnitude of the dust forcing also determined the phase of the glacial cycle during which the AMOC is most sensitive to radiative forcing: pronounced reductions in radiative forcing under strong scaling resulted in a shift to the weakest AMOC mode early in the last glacial cycle, which is from then on insensitive to further changes induced by additional reductions in radiative forcing later on. Conversely, under weaker scaling, the

initial decrease in forcing was insufficient to shift the AMOC out of its interglacial circulation
mode.

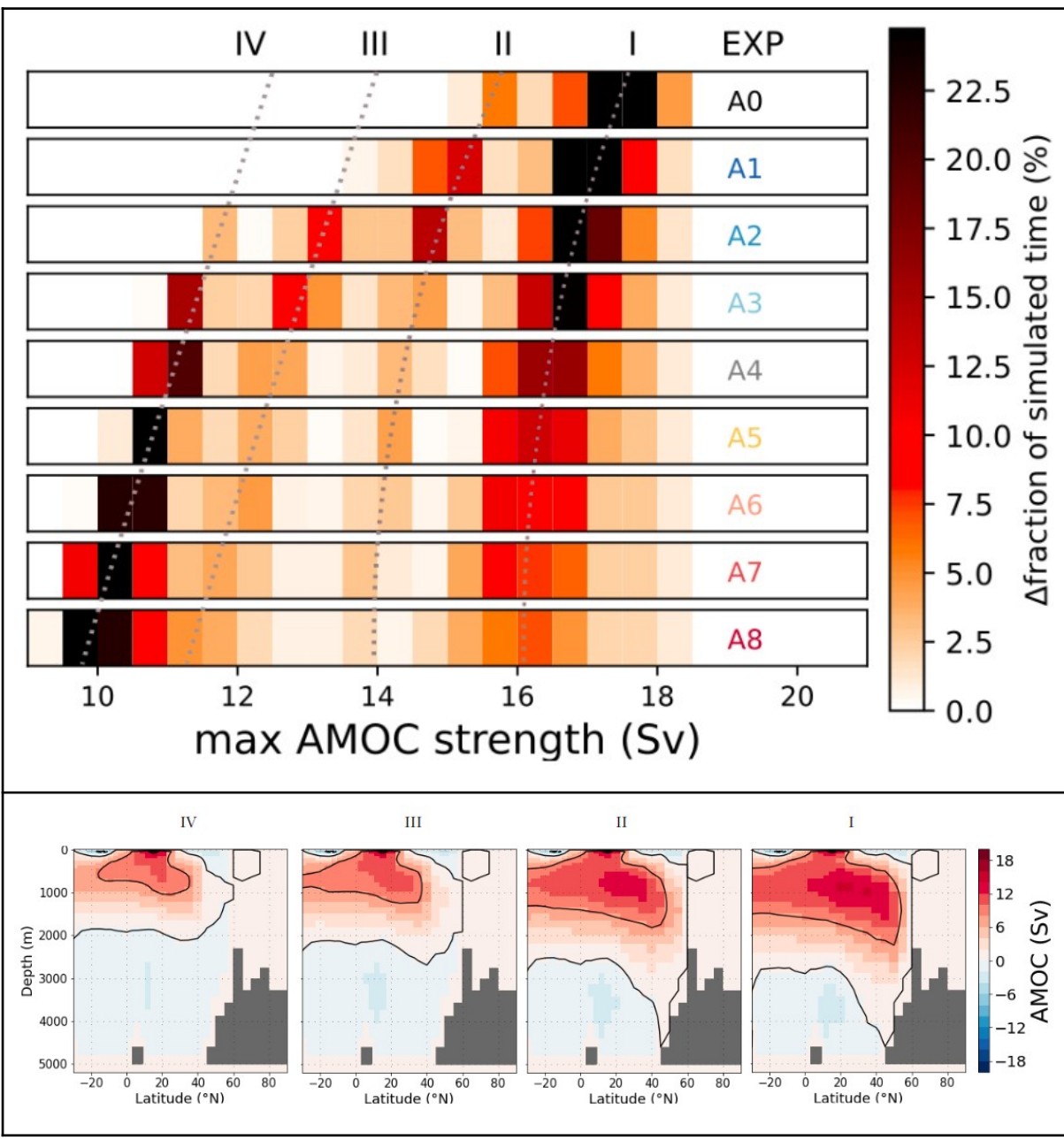

Figure 2: Top: Fraction of each simulation in simulation set A (each over 788 kyr) during
which a given maximum AMOC strength was simulated. Each row shows the results of one
simulation, with the simulation ID on the right end of the column in colours that correspond to
the lines in Fig 1. The bins are 0.5 Sv wide and four relative maxima in occurrence,
exhibiting distinct AMOC modes, $I - IV$, are indicated by dotted lines. Bottom: AMOC stream
function for the four circulation modes adopted across the last glacial cycle in simulation A3.

All simulations revealed multiple intermediate circulation modes between the glacial and
interglacial end-members. These modes manifested as distinct bands of increased
occurrence in Fig 2, which displays the fraction of the entire simulated period of 788 kyr
during which the AMOC exhibited a given maximum strength (binned into 0.5 Sv intervals).
The two intermediate modes II and III are distinguishable by AMOC strength, but not by their
meridional temperature or salinity gradients (Fig. SI.4), which questions whether these are
indeed separate circulation modes or expressions of one single mode that can have different
AMOC strengths (Lohmann et al. 2023). Yet, these circulation modes differ in global mean
and Greenland temperatures and North Atlantic Sea ice cover, suggesting that they are still
separate climate states (Fig. SI.5). Thus, we identified four frequently occurring circulation
modes in simulation set A that can be distinguished by AMOC strength, sea ice and
temperature, and three which can be distinguished by meridional temperature and salinity
gradients.

AMOC transitioned between these modes across the simulated glacial cycles due to
radiative forcing (Fig 2). The glacial and interglacial 'end-member' circulation modes I and IV
occured most commonly: The AMOC was in either of these two modes for 62-85% of the
simulated 788 kyr, depending on the dust forcing scaling. The AMOC was found in the
intermediate circulation modes II and III most commonly under weak dust forcing. For
stronger forcings, AMOC transitioned quickly through these modes, which were therefore
less frequently occupied. Thus, it appears that there is a tendency towards bi-modal AMOC
stability under strong forcing scaling, where the AMOC was almost exclusively either in the
glacial or interglacial circulation mode. Once AMOC had adopted the weakest mode,
additional reductions in radiative forcing only caused minor additional and gradual AMOC
weakening and did not cause another abrupt transition.

The simulations A3 and A4 with intermediate glacial-interglacial temperature changes (LGM-
PI ΔGMST -5 to -6 °C, similar to the -6.1 °C constrained by Tierney et al., 2020)
predominantly exhibited AMOC transitions between the interglacial (mode I, ~16-17 Sv) and
glacial mode (mode IV, ~11 Sv), with two rarer intermediate circulation modes in-between.


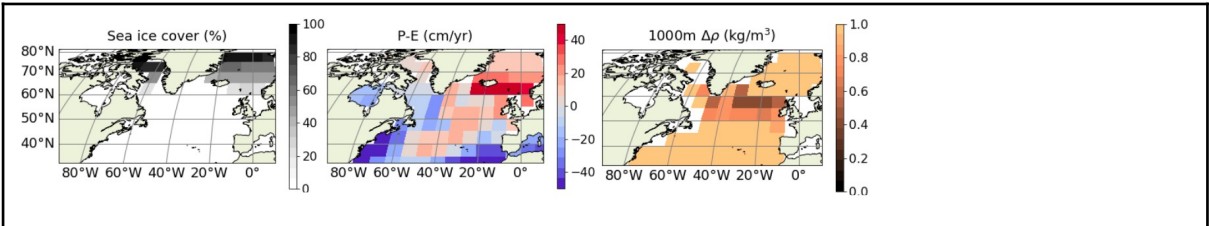

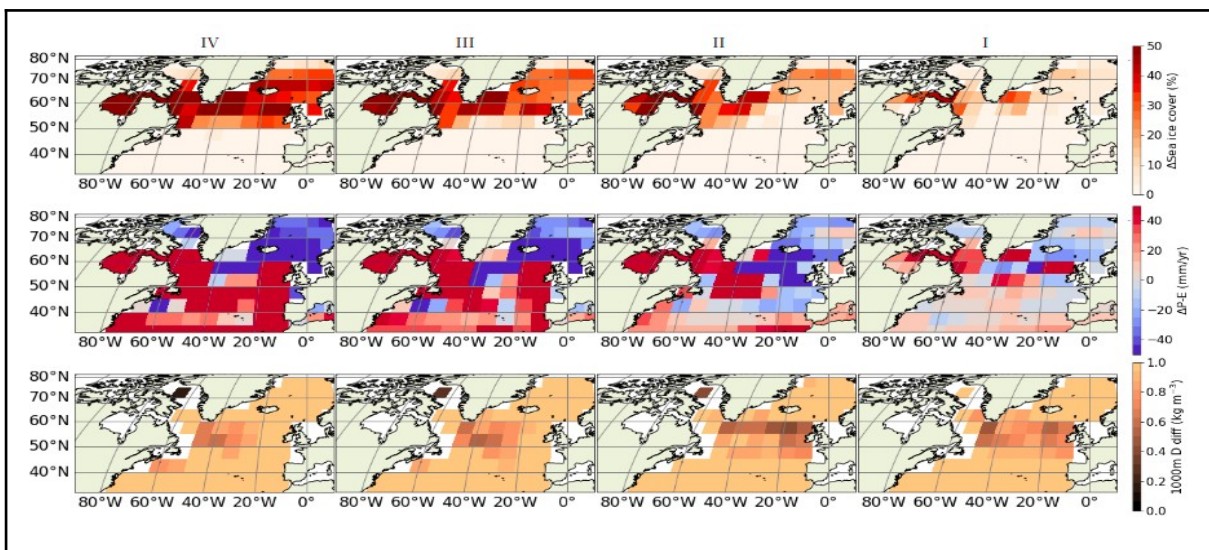

Figure 3: Top row: Initial annually averaged sea ice cover, meteoric freshwater balance, and the density difference over the uppermost 1000 m of the water column in the North Atlantic . Panels below: Differences relative to the initial state for annually averaged sea ice cover, meteoric freshwater balance, and the density difference over the uppermost 1000 m of the water column in the four circulation modes.

The interglacial circulation mode (mode I in Figs. 2 and 3) is characterised by NADW formation in the subpolar North Atlantic, specifically south of Greenland and close to the British Isles, as indicated by the small density difference over the upper 1000 m of the water column. In the first intermediate AMOC mode (II), deep water formation is enhanced in the Eastern Atlantic while it weakens in the West as sea ice expands further South (Fig. 3). The next intermediate circulation mode (III) is marked by a reduction in deep water formation in the eastern North Atlantic, as the local water column increasingly stratifies. Deep water formation continues south of the sea ice edge in the western North Atlantic, albeit substantially weakened. As the northwards transport of subtropical water diminished under further cooling, the AMOC transitioned into the glacial stable mode (IV). In this mode, convection in the North Atlantic is strongly reduced and cold, fresh surface waters stratify the water column off the European coast. At this point, additional negative radiative forcing enhanced the amplitude of the temperature and salinity anomalies but without triggering additional changes in the North Atlantic circulation pattern.

Our simulations cover four glacial cycles before the Mid-Brunhes transition (MBT, MIS 12 and MIS 11 (~430 ka)) and four thereafter. This transition was marked by a shift to warmer interglacials with higher atmospheric $CO_2$ concentrations. There are only small differences between the distributions of AMOC modes before and after the transition (fig SI.2), and none

are statistically significant in the two-sided Smirnov test, which determines the likelihood that
two distributions are the same (Berger and Zhou, 2014), even at the 50% confidence level.

**3.2. Processes responsible for the AMOC changes**

a)  b)

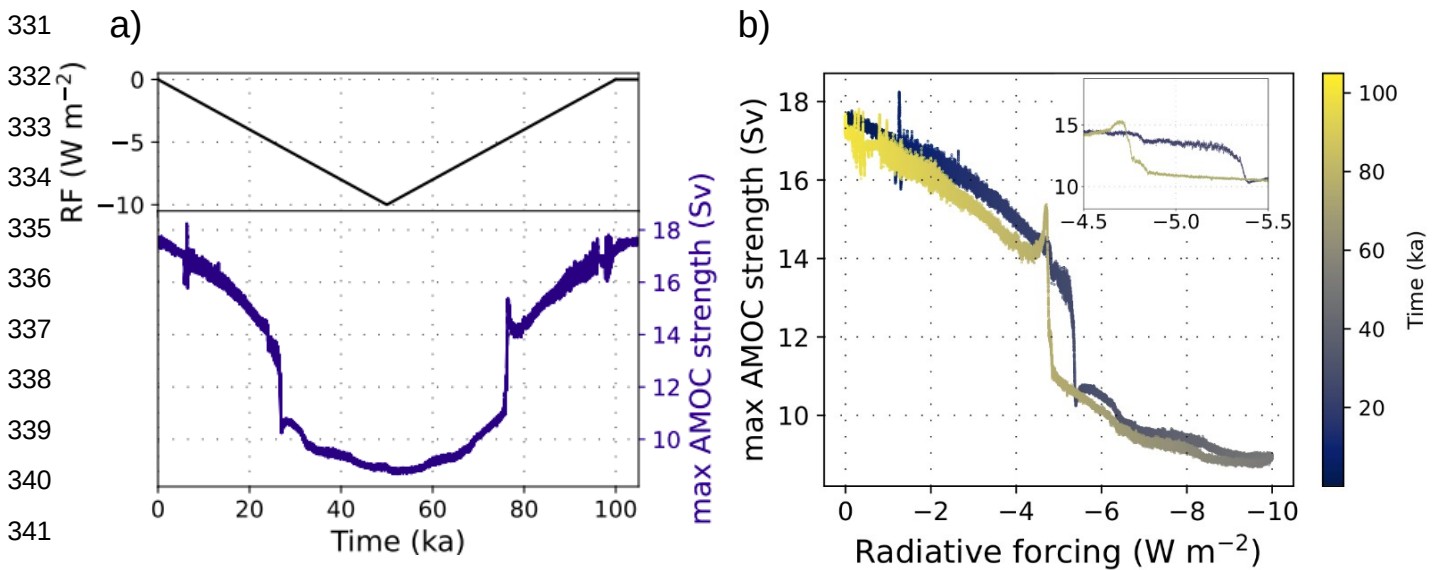

Figure 4: Simulation B.slow: (a) Response of the AMOC to changes in radiative forcing
relative to the pre-industrial. The radiative forcing was linearly decreased over 50 kyr to a
minimum of -10 W/m$^2$ and then increased again at the same rate. (b) The associated
hysteresis loop of the AMOC under the radiative forcing, with the inset providing an enlarged
view of the hysteresis loop.







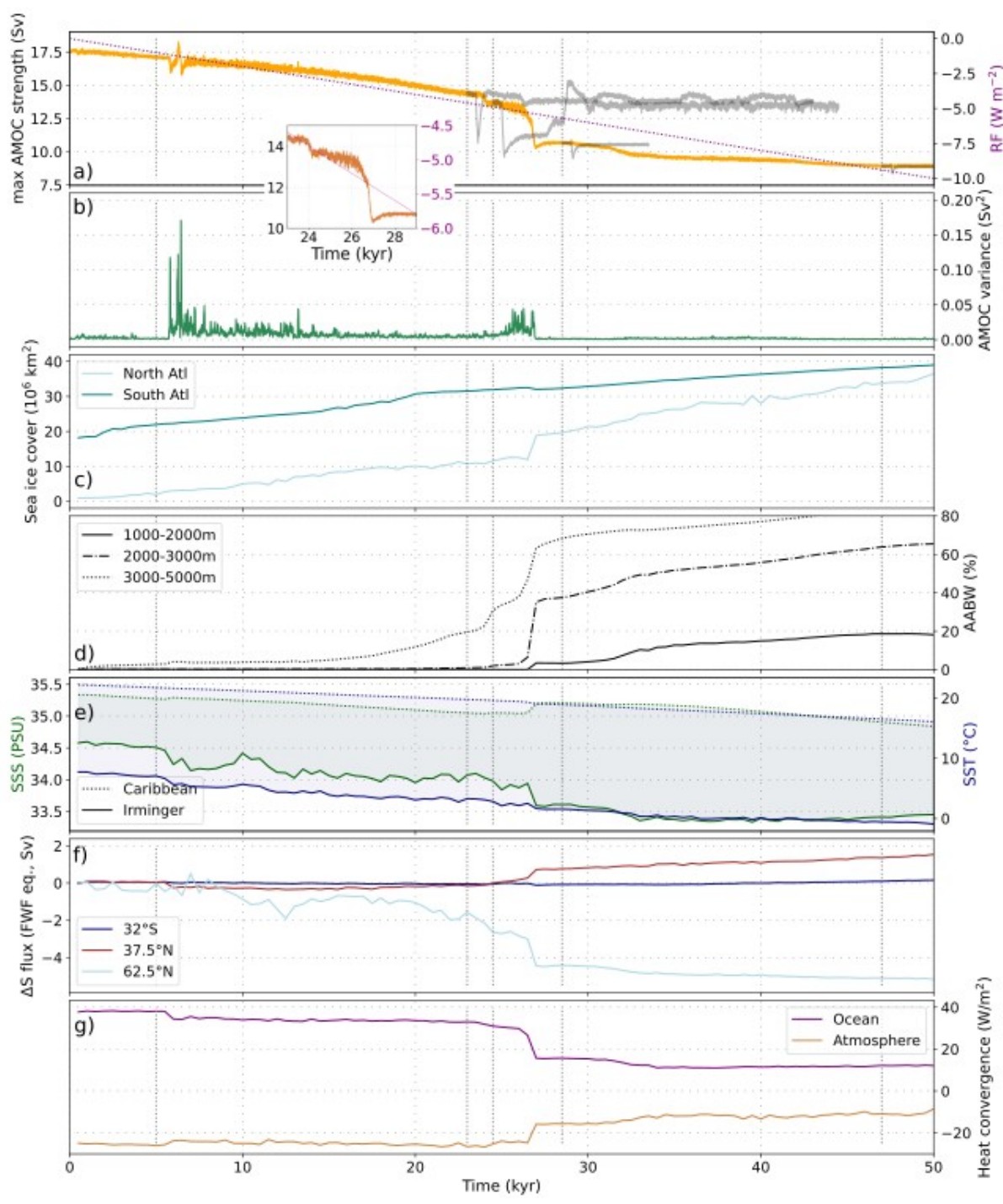


Figure 5: Changes in ocean properties during the cooling phase in simulation B.slow. a)
AMOC strength and the applied radiative forcing. At four points in time throughout B.slow,
simulations were branched off to test the stability of the respective circulation mode (shown
in dark grey). In these simulations, we kept the radiative forcing constant but applied a small
freshwater perturbation after 500 yrs, before allowing the model to re-equilibrate (see
Methods). b) AMOC variance calculated in a 50 yr moving window. c) Sea ice cover in the
North Atlantic between 50-60°N ('North Atl', light blue) and the Atlantic sector of the
Southern Ocean 50-68°S ('South Atl', teal). d) Volume fraction of AABW at three different

depth intervals in the subpolar North Atlantic (50-60°N). e) SST and SSS in the Caribbean and Irminger seas. f) Change in the northward salinity transport by ocean currents in freshwater flux (FWF) equivalents at different latitudes (following Liu et al., 2017). g) Column-integrated heat flux convergence due to ocean circulation and heat loss to the atmosphere (negative = heat loss by ocean) for the North Atlantic (40°N-70°N). Dotted vertical grey lines indicate time points in the simulation at which we branched off stability tests, and at which we analysed water mass distributions in Fig. 6.

In our simulations, the primary processes controlling the AMOC strength under changing radiative forcing are density changes due to heat and salinity redistributions. We investigated these in more detail in experiment B.slow (Fig. 4 and 5). This experiment is characterised by a slow linear decrease in radiative forcing over 50 kyr, before it is increased again to the pre-industrial value with the same rate of change (Fig. 4). Fig. 5 shows that AMOC weakened gradually over the first 24 kyr, then weakened abruptly by 1 Sv at 24 kyr into the simulation and by ~3 Sv at 27 kyr, and then continued to weaken gradually until the forcing is reversed (Fig. 5a). In addition to the abrupt transition in AMOC strength, we found several additional rapid changes in AMOC variability, heat, and salt fluxes (Fig. 5) and regional density profiles (Fig. SI.7-9) which were not associated with persistent changes in AMOC strength, e.g. at 6 kyr into the simulation. In fact, experiment B.slow shows that a cascade of changes with little effect on the mean AMOC strength occurred before the first abrupt AMOC weakening after 24 kyr. Since these changes might partially be artifacts of our coarse model resolution, we here only focus on the larger scale changes instead. Initially, the whole Atlantic surface ocean cooled and freshened, leaving the temperature and salinity differences between the Irminger and Caribbean Seas almost unchanged (Fig 5e). However, NADW became less salty and colder as a consequence of the changes in the surface ocean (not shown) and the vertical density profiles in the subpolar North Atlantic steepened due to the surface freshening and deep ocean cooling(Fig. SI.7-8).

After about 6 kyr, NADW formation moved south as surface freshening stabilised vertical density profiles in the subpolar east North Atlantic and density profiles further south steepened due to surface cooling combined with subsurface warming (Fig. SI.7-9). These changes did not cause a step-change in AMOC strength, but freshwater and heat advection into the North Atlantic was reduced(Fig. 5f, g), which reduced North Atlantic SST and SSS (Fig. 5e). Sea ice expansion increased in the eastern North Atlantic, and AMOC variance (calculated over a moving 50-year window) was increased (Fig. 5). The reduced influx of subtropical surface waters also caused abrupt cooling and freshening in the Irminger Sea (Fig. SI.8). At 24 kyr, the AMOC had weakened to ~14.5 Sv and sea ice cover extended

south of the Irminger Sea (Fig SI.11). At this point, the AMOC strength dropped abruptly by 1
Sv, and then by an additional 3 Sv ~3 kyr later, as the reduced salinity advection into the
North Atlantic and a net increase in precipitation minus evaporation (P-E) led to a strong
surface freshening. As a result of the North Atlantic density changes, the main North Atlantic
convection site shifted southwards (determined by changes in the vertical density profiles,
Fig SI.10). Sea ice also increasingly covered former areas of deep water formation in the
North Atlantic. In the weakest circulation mode, the location of the maximum AMOC stream
function shifted southwards by approximately 10 degrees and up in the water column by 400
m initially (28.5 kyr) and eventually almost 800 m (47 kyr) This shift allowed cold, less dense
water masses to extend further south into the North Atlantic.

In the Southern Ocean, the cooling enhanced Southern Ocean deep water formation early
on in the experiment and led to a continuous expansion of sea ice in the Southern
Hemisphere. The biggest AMOC weakening at ~27 kyr was also accompanied by a very
weak bipolar seesaw effect (Stocker and Johnsen, 2003), which caused a temporary decline
in sea ice coverage in the Atlantic sector of the Southern Ocean (Fig. 5). This sea ice
decline, however, was too small to reduce the radiation-driven sea ice increase in the longer
term. Both shifts in AMOC strength were accompanied by an increased spread of AABW into
the North Atlantic (Fig. 5d). The volume of AABW in the deep Atlantic influences AMOC
stability (Zhang et al., 2013, Galbraith and Lavergne, 2019). Thus, the spread of AABW into
the deep North Atlantic after the first AMOC shift at ~24 kyr might have preconditioned the
AMOC for the following shift at ~27 kyr in B.slow.

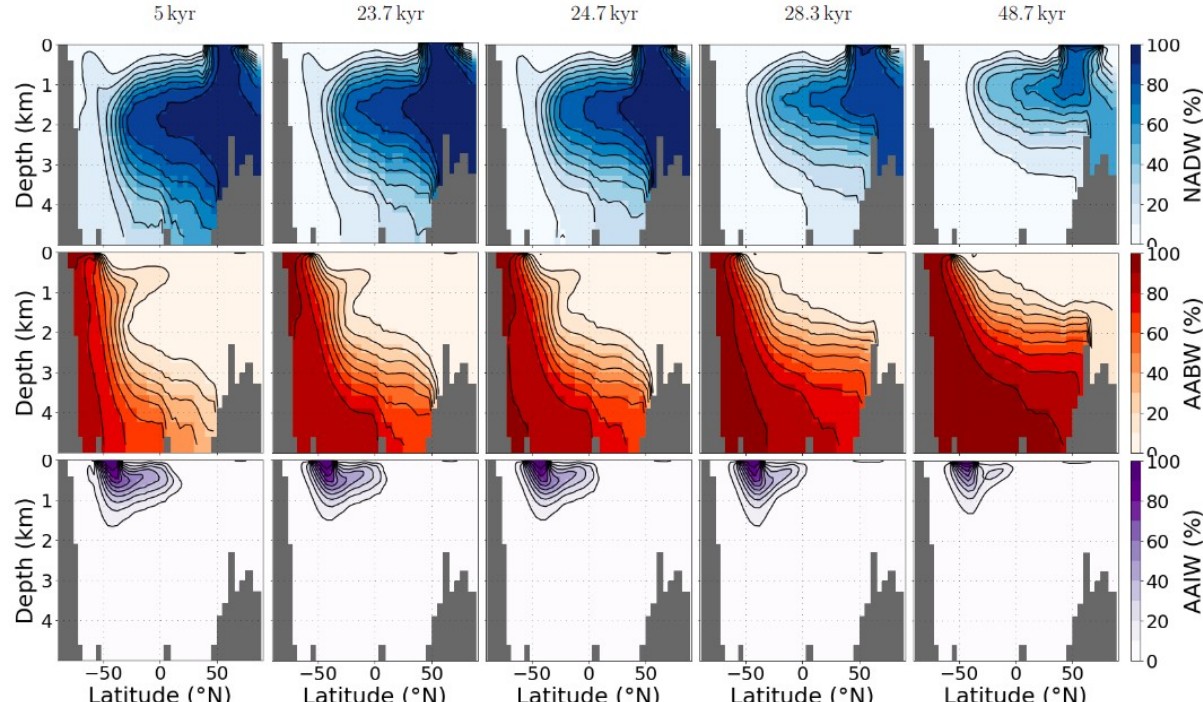

Figure 6: Atlantic water mass distributions at the five time slices of our simulation B.slow indicated in Fig. 5. Each row shows the zonally-averaged contribution of water sourced in one of three regions: the North Atlantic (upper row), the Southern Ocean (middle row), and the Southern Atlantic (bottom row), diagnosed with three passive dye tracers. Fig. SI.1 shows the spatial pattern of our dye forcing.

The changes in the AMOC stream function associated with the decreasing radiative forcing in experiment B.slow bear close resemblance to the changes we observed in the transient experiment set A during AMOC transitions from the interglacial to the glacial circulation mode (Fig. 6 and Fig. SI.12 - SI.14).

We tracked the effects of these circulation changes on the Atlantic distribution of intermediate and deep water masses as diagnosed from artificial dye tracers (see Fig SI.1 for their source regions). Figure 6 shows that, during the first 23 kyr of our simulation, AABW slowly spread further North and occupied increasingly shallower depths while the northward reach of AAIW was reduced. Accordingly, NADW shoaled as it was unable to sink further when encountering AABW in the deep North Atlantic. The reduced export of NADW also led to a decrease in its southward extent, contracting to 40°S. The first abrupt shift in AMOC strength occurred at 24.5 kyr in B.slow and had only small effects on the water mass distribution. It mainly led to a reduced fraction of NADW at intermediate depths of the North Atlantic >45°N and a small increase of AABW in the abyssal North Atlantic (Fig. 5d). The following AMOC shift at 27 kyr reduced AMOC strength by more than 3 Sv, and was hence

also more strongly expressed in changes in the water mass distribution. It was accompanied
by a further reduction of NADW export into the deep Atlantic, before NADW was entirely
replaced by AABW at depths below ~3.5 km in the weakest circulation mode. AAIW was
increasingly curtailed in its northward reach, until it effectively no longer extended toward the
equator (<10%).

In summary, in our simulation deep convection diminished first in the Irminger Sea while
deep water formation continued in the subpolar Northeast Atlantic and south of Greenland.
As sea ice extended into the Eastern North Atlantic south of Greenland and vertical density
profiles steepened further south, the northward reach of the AMOC was restricted and a new
circulation mode was established with increased sea ice cover >55°N. The weakened
northwestward transport of heat and salt due to the reduced AMOC strength led to a
relatively fresh and cold eastern North Atlantic, stabilising the water column in the region and
producing another persistent AMOC mode. The simulated step changes in AMOC strength
in our simulations were thus the response to gradual surface cooling and freshening, and
occurred when NADW formation shifted southwards. The resulting redistributions of heat
and salinity caused sudden shifts in the vertical density profiles and sea ice expansion which
consolidated the new circulation mode (Ando and Oka, 2021). In particular, reduced
advection of heat and salinity into former locations of deep water formation resulted in a
more stable local water column (Fig. SI.7-9). The deep water formation regions are sensitive
to heat and salt flux changes, because any reduction in sea surface temperatures (SST)
increases surface density but simultaneously reduces evaporation in ice-free areas, thus
effectively creating a small freshwater forcing and a negative feedback to the buoyancy
changes caused by the initial SST decrease. Sea ice covering the downwelling areas
stabilises the water column by preventing surface ocean cooling and evaporation. The
progressive influx of AABW into the North Atlantic is a further process stabilising new
circulation modes by stratifying the water column from below (Buizert and Schmittner, 2015).
The difference between freshwater transport into the South Atlantic at 32°S and into the
Arctic at 62.5°N in Fig. 5f can be used as a measure for the basin-wide salinity feedback
(Rahmstorf, 1996, de Vries and Weber, 2005). In our simulation, changes in this metric were
predominantly caused by changes in the transport across the northern edge, since transport
into the South Atlantic remained almost unchanged throughout the cooling phase of B.slow.
North Atlantic salinity is instead governed by changing transport from the subtropics into the
North Atlantic and between the North Atlantic and Arctic. As such, in our simulations it
seems the processes involved in the sudden AMOC strength changes, namely density
changes in the upper water column, and those that stabilised new circulation modes (salinity
and heat redistributions, sea ice expansion) mostly operated in the North Atlantic region.
Our stability experiments demonstrated that the circulation modes before and after the
abrupt shifts recovered from small freshwater perturbations, and can thus be considered
stable, i.e. sufficiently far from bifurcation points to recover from the small perturbation (Fig.
5a, Fig. SI.6). In these branched off sensitivity tests, the circulation mode adopted before the
first AMOC threshold (at ~24 kyr), showed increased variability in the order of 0.5 Sv. The
next circulation mode (~25 kyr) responded most strongly to small freshwater perturbations
and was also the only circulation mode in our simulation which showed gradually increasing
AMOC variability (as determined by an increase in its variance) while approaching the next
threshold (Fig. 5a, Fig. SI.6). When the forcing was reversed, the radiation increase
gradually strengthened the AMOC until it rapidly transitioned back into the stronger
circulation mode when North Atlantic sea ice had receded sufficiently for a northward shift of
the convection sites and evaporation and salinity transport resumed. The radiative forcing at
which the AMOC transitioned from one circulation mode to the other was not equal for
decreasing and increasing radiative forcing: a stronger negative radiative forcing was
required to push the AMOC into its weak circulation mode than for the transition out of it (Fig.
4b).
Our sensitivity tests with different orbital configurations indicated that the existence of AMOC
thresholds under radiative forcing was not dependent on the initial orbital configuration.
However, the AMOC was slightly more sensitive to perturbations when initiated with the
orbital configuration equivalent to 30 ka before  present. In this case, the threshold for the
AMOC to transition to its weaker mode was reached ~1 kyr earlier than under PI or 50 ka
orbital configurations (simulations B.short.30ka, B.short.PI, Fig. SI.15). The processes that
affected AMOC behaviour in simulation set B also caused AMOC changes over the
transiently simulated 788 kyr in simulation set A, but the circulation modes adopted varied
slightly in sea ice extent, hydrological cycle and salinity distribution under varying orbital
configurations.
**3.3. Comparison with other modelling studies and proxy data**
In our transient simulations covering the past 788 kyr, the AMOC strength decreased during
glacial phases solely due to changes in the hydrological cycle and sea ice that were induced
by orbital, greenhouse gas, and the additional radiative cooling. The existence of multiple
stable AMOC modes under varying thermal or radiative forcings has been found in various
GCMs (e.g. Knorr and Lohmann, 2007, Oka et al., 2012, Banderas et al., 2012, Brown and
Galbraith, 2016, Zhang et al., 2017, Klockmann et al., 2018). In agreement with previous
studies, we found multiple persistent AMOC circulation modes with distinct AMOC strengths
for radiative forcing levels between full glacial and interglacial climate states. Moreover, we
found that the transitions between these modes occur abruptly, some within as little as 100
years. In accordance with Lohmann et al. (2023), we found that these shifts in AMOC
strengths are preceded by cascades of density and circulation field changes, the number
and sequence of which depend on the strength of the forcing. Similar to the findings from
Oka et al. (2021), AMOC transitions arise primarily from salt redistribution in the ocean and
sea ice expansion into deep convection zones.

In our simulations A and B, each transition in AMOC strength was associated with a shift in
the convergence of heat and salt fluxes and a southward expansion of sea ice into the North
Atlantic. Sea ice cover decouples the surface ocean buoyancy from the atmosphere. In the
intermediate modes, locations with steep density gradients are close to a critical annually-
averaged sea ice cover. In these modes, small changes in sea ice cover can cause large
changes in surface buoyancy and the extent and location of deep convection, which makes
the AMOC sensitive to small perturbations. The AMOC was only pushed into its weakest
mode when all former convection sites in the subpolar North Atlantic were sea ice-covered
and heat convergence in the North Atlantic was strongly reduced.

In their examination of thermal forcing of both hemispheres in COCO, the ocean component
of MIROC, Oka et al. (2021) found that thermal AMOC thresholds only exist if the Southern
Hemisphere is cooled more than the Northern Hemisphere. In contrast, Zhang et al. (2017)
found sudden AMOC changes due to greenhouse gas changes without a special focus on
the Southern Hemisphere. In our simulations with Bern3D, we also found thermal thresholds
with similar cooling rates in both hemispheres, but only after salinity re-distributions and
changing meteoric freshwater fluxes in response to about six thousand years of global
cooling. Thus, in our model, Southern Hemisphere cooling does not need to exceed the
cooling of the Northern Hemisphere to affect AMOC but further sensitivity tests would be
required to establish the relevance of cooling in each hemisphere separately (as shown in
Oka et al., 2021).

It is possible that changing meteoric freshwater fluxes are essential for the existence of such
a thermal threshold, which does not therefore appear in COCO without a thermally
responsive atmosphere with a climate-driven freshwater balance. In a model with a dynamic
energy moisture balance component, atmospheric cooling reduces evaporation and the
water-holding capacity of the atmosphere. With this feedback enabled in our model, cooling
can then affect seawater density directly via changing temperatures, and indirectly via
changing the meteoric freshwater balance and surface salinities. These changes would
induce additional kinematic changes (i.e., in the wind fields) in fully dynamic atmosphere
models but are kept constant in our simulations, i.e. in our simulations the moisture content
of air changes with climate but not the direction or strength of winds which disperse it. A
decrease in the water-holding capacity of air therefore directly leads to a reduction of the
large-scale atmospheric moisture transport from low to high latitudes.


The primary importance of salinity and heat redistributions as well as sea ice extent in the
North Atlantic for the simulated AMOC shifts resembles the findings from Ando and Oka
(2021)'s hosing experiments under LGM conditions and Zhang et al. (2017)'s simulations of
AMOC shifts in response to $CO_2$ changes under intermediate-glacial conditions. While our
experiments were run with pre-industrial topography, sea level and wind fields, the initial
location of convection sites between Greenland and the British Isles (areas with lowest
density differences over upper 1000 m in Fig. SI.11) resembles the LGM and intermediate-
glacial circulation modes in Ando and Oka (2021) and Zhang et al. (2017).

Ganopolski and Rahmstorf (2001) found that the possibility of a southward shift of deep
convection depends on the latitude of prior deep convection and the density field further
south, and Oka et al. (2012) showed that the location of deep convection and its distance
from the winter sea ice edge define thermal thresholds in AMOC strength. Several controls
on the location and strength of deep convection in the North Atlantic, that would have
affected AMOC stability over glacial cycles, have been established. Changes in wind stress,
for example, have been documented to exert important controls on AMOC stability (e.g.
Arzel et al., 2008, Yang et al., 2016) and thermal thresholds (Oka et al., 2012), but in our
simulations wind stress is constant. Besides wind fields, the location of deep convection is
further dependent on climate and sea level/bathymetry (Ganopolski and Rahmstorf, 2001,
Oka et al., 2012, Zhang et al., 2014b, Zhang et al., 2017), and thus the thermal AMOC
thresholds are model and forcing dependent (Oka et al., 2012). Our simulations capture the
albedo effect of varying terrestrial ice sheet extent, but we did not consider their orography
or sea level effects, including impacts on the atmospheric circulation, which were shown to
affect AMOC (Li and Born, 2019; Pöppelmeier et al., 2021). Previous studies suggested that
pre-industrial or intermediate glacial ice sheet configurations are required to even produce a
thermal AMOC threshold in the range of glacial-interglacial $CO_2$ concentrations in a full GCM
and that the presence of a full glacial Laurentide ice sheet prevents such a threshold (e.g.
Klockmann et al., 2018). Northern Hemisphere ice sheets also affect the composition and
volume of AABW through teleconnections (Galbraith and Lavergne, 2019), and the
buoyancy difference between AABW and NADW, as well as their fraction in Atlantic deep
water, have been found to precondition AMOC stability (Zhang et al., 2013). In addition,
changes in the interconnection of marine basins, specifically the Bering Strait, also affect
AMOC stability (Hu et al. 2012). The values of the thermal thresholds in our experiments are
thus likely sensitive to the model design and initiation. Pöppelmeier et al. (2021) showed that
the sensitivity of Bern3D to freshwater hosing increases when additional LGM boundary
conditions are prescribed (changed wind fields, closed Bering Strait, tidal mixing differences
due to sea level changes). The different wind fields and tidal mixing strengthened AMOC
and increased the salt and heat transport into the subpolar North Atlantic. This could mean
that stronger cooling is required to stabilise the water column in the Irminger Sea and reach
the first thermal threshold, when the full range of glacial boundary conditions are applied.
Closure of the Bering Strait increased the salt advection feedback, which stabilises the weak
circulation state without deep water formation in the subpolar North Atlantic.

Further investigations are needed to determine how changes in strength and location of the
wind stress due to the ice sheet's orography, sea level and Bering Strait closure would affect
sea ice formation in the northern North Atlantic and the AMOC thresholds in our simulations
quantitatively. Since we chose to focus only on radiation driven AMOC changes in our
experiments, while in reality AMOC was also influenced by freshwater flux changes,
particularly during Heinrich events, we would not expect a close model-data match with
reconstructed millennial-scale AMOC changes in the paleo-records. Still, we can compare
the long-term evolution of AMOC strength in our simulations and the reconstructions. Our
simulations show that the reconstructed glacial-interglacial temperature changes had the
potential to alter the density field in the North Atlantic by redistributing heat and salt, and that
some of these changes might have resulted in abrupt changes of AMOC strength. By testing
a wide range of glacial-interglacial temperature changes, our experiments demonstrate that
the cooling during glacial periods likely contributed to a weakened AMOC. The strength and
timing of the weakening depends on the actual temperature change in the North Atlantic
which would have been modulated by changes in winds and ice shields.

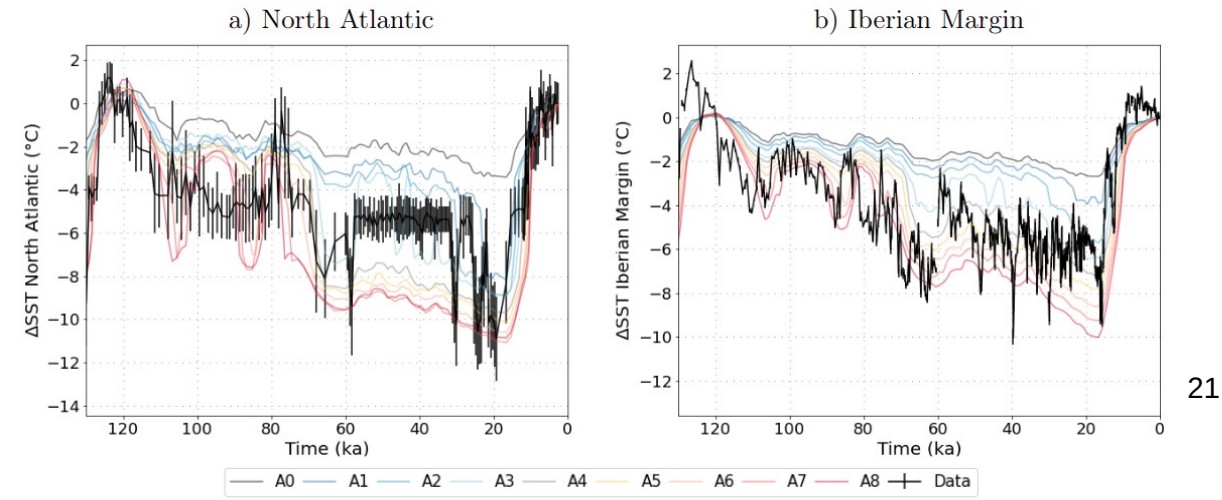

Figure 7: Simulated and reconstructed SST differences from PI over the last glacial cycle in the North Atlantic (a, reconstruction by Candy and Alonso-Garcia, 2018) and on the Iberian Margin (b, reconstruction by Davtian and Bard, 2023). The model data was interpolated to the time points for which proxy reconstructions exist.

Unlike in our simulations, most GCMs participating in PMIP4 do not show a shoaling or weakening of the overturning cell under LGM boundary conditions (Sherriff-Tadano and Klockmann, 2021). The difference could arise from the static wind fields that we prescribed, since an ice-sheet related increase in wind speeds over the North Atlantic leads to a strengthened AMOC (Klockmann et al., 2018), or different representations of processes affecting AABW density changes (e.g. brine rejection, Bouttes et al., 2011). A shallower and likely weaker AMOC during peak glacials is however consistent with observational data (Lynch-Stieglitz et al., 2017, Pöppelmeier et al., 2023). In Fig. 7, simulated SST changes from the Rockall Trough and the Iberian Margin are compared to proxy-based reconstructions. Circulation changes alter the distribution of heat in the North Atlantic, and simulated SST patterns are strongly affected by AMOC changes. In response to the stepwise AMOC weakening, simulated Atlantic SST also transitioned stepwise from interglacials to glacial maxima. Step changes are also an established feature of Atlantic SST reconstructions over the last glacial cycle (Fig. 7), with the biggest steps at 120-110 ka and 80-60 ka also captured in our simulations. During glacial inception between 120 ka and 70 ka, the amplitudes of reconstructed SST changes in both locations resemble those simulated with strong radiative forcing (simulations A6, A7, A8). Afterwards, SSTs in those simulations decreased more than in the reconstructions, and the latter align more closely with weaker radiative forcing (simulations A3, A4). After ~70 ka, shorter millennial-scale events (Heinrich and Dansgaard-Oeschger), that were not included in our simulations, were more frequent than before and could affect the comparability between reconstructed and simulated SST. Additionally, the further into the glacial cycle, the more the topography and wind fields would have deviated from their pre-industrial states that we kept constant throughout the simulations. These factors could have caused a shift in AMOC and SST changes that are not captured by our simulations.

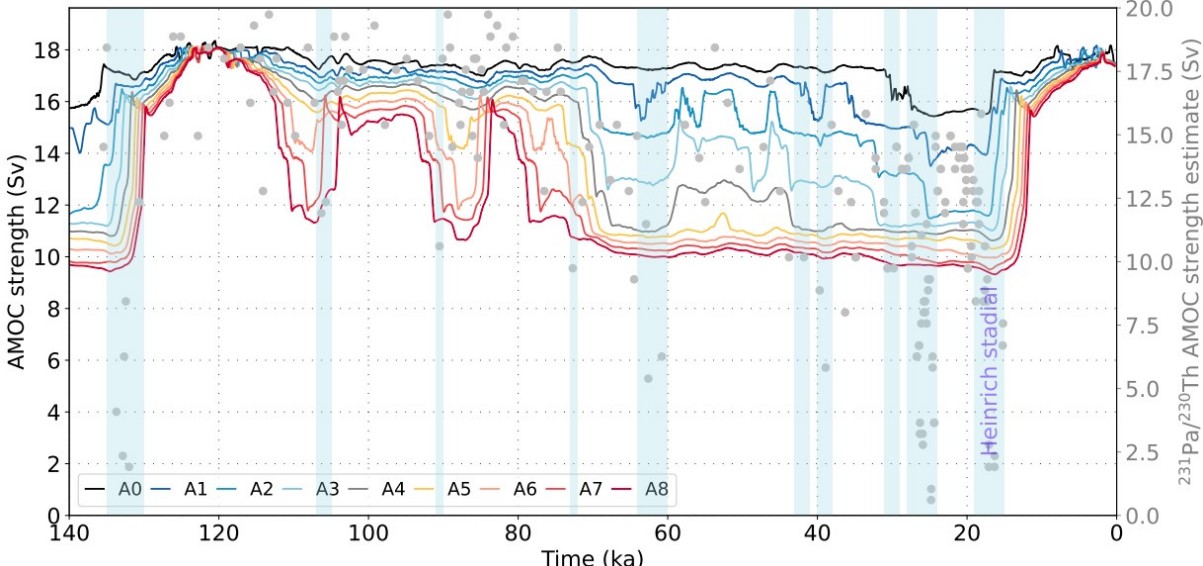

663

Figure 8: Simulated AMOC changes due to thermal forcing over the last 140 kyr. Gray dots indicate AMOC strength estimated from $^{231}$Pa/$^{230}$Th (Böhm et al., 2015, Lippold et al., 2009) by assuming a sensitivity of -0.0024 Sv$^{-1}$ (Rempfer et al., 2017).

Fig. 8 compares the simulated changes in AMOC strength over the last 120 kyr in simulation set A to indications of AMOC weakening based on $^{231}$Pa/$^{230}$Th from the Bermuda Rise (Böhm et al., 2015). The simulations A2-A4 have PI-LGM GMST differences of 4.7-6.2°C (within the proxy-constrained and PMIP range and close to the most recent estimate of 6.1°C by Tierney et al., 2020) and show a shift to a weaker AMOC at the beginning of MIS 4 around 70 ka ago, when a negative $^{231}$Pa/$^{230}$Th shift occurred. While the simulated radiation-driven AMOC changes cannot explain weaker or collapsed circulation modes (<11 Sv) during Heinrich stadials, this comparison shows that the long term AMOC weakening during glacial phases could have been driven by temperature changes. It is important to note that AMOC strength estimates based on this $^{231}$Pa/$^{230}$Th record need to be treated with caution. Pöppelmeier et al. (2021; 2023) showed a strong local influence on sedimentary proxies at this site, and we did not correct the $^{231}$Pa/$^{230}$Th signal for potential productivity changes.

**3.4. Meta-stable AMOC modes over the last 788 kyr**

Finally, we can test whether our simulations capture the periods with increased frequency of AMOC transitions that are indicated by proxies over the last eight glacial cycles. Using our 788 kyr long simulations in simulation set A, we determined how often and when the radiative forcing pushed the AMOC into 'excitable' circulation modes, i.e. modes II and III, which showed more frequent AMOC strength shifts than the interglacial and glacial modes I and IV (Fig. 1 and SI.2), and how this varied with the applied forcing strength (Fig. 9). In all

simulations, the AMOC transitioned into such excitable modes in all of the past eight glacial
cycles, but the timing of these shifts varied. For example, during the last glacial cycle, the
simulations A2-A4 exhibited an intermediate circulation mode during MIS 3 (57-29 ka), when
frequent AMOC mode shifts occurred (see Fig. 1). Similar rapid mode switches occurred
earlier in the glacial cycle, i.e. during MIS 5d-e in simulations A6-A8. In these simulations,
the AMOC already transitioned into the persistent glacial circulation mode IV at the
beginning of MIS 4 (71-57 ka), in which  North Atlantic density profiles are more stable. In
simulations A1-A3, the AMOC persisted in these modes for several tens of thousands of
years at a time, during most glacials. Under stronger radiative forcing, the periods in which
AMOC adopted these modes were shorter and mostly occurred at the start of glacial cycles.

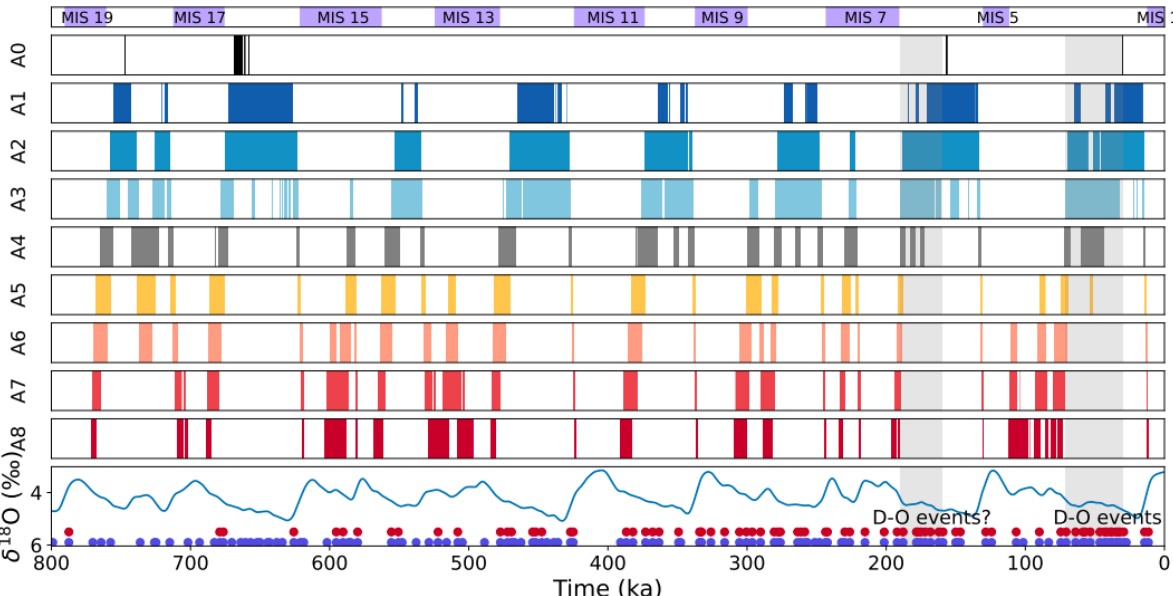


Figure 9: Occurrence of intermediate AMOC modes II and III due to radiative forcing over the
last 788 kyr in simulation set A. The time periods with intermediate AMOC modes are
marked as vertical bars, each row showing the results for a different forcing magnitude from
simulation set A. At the bottom, $\delta^{18}$O from Lisiecki and Raymo (2005) is shown for reference,
alongside the time period with confirmed and suspected Dansgaard-Oeschger events (light
gray bars based on Rousseau et al., 2020, blue and red circles are based on reconstructions
Barker et al., 2011, who used two different detection thresholds). The gray bars indicate the
periods in MIS3-4 and MIS6 with confirmed Dansgaard-Oeschger events.

We can assess the skill of our simulations at predicting 'excitable' AMOC modes from the
radiative forcing by comparing the output with records of high AMOC variability in the past.
Simulations A3 and A4 shift into a meta-stable circulation mode during MIS 3, and similarly
between 190 and 160 ka during the penultimate glacial cycle, and prior to each previous

glacial maximum but not during the glacial maxima themselves. An 'excitable' AMOC mode during these intervals seems realistic given the high frequency of Dansgaard-Oeschger events in MIS 3 and the suspected occurrence of Dansgaard-Oeschger events during MIS 6 (191-123 ka, Rousseau et al. 2020). Similarly, Barker et al. (2011), who predicted the occurrence of Dansgaard-Oeschger events during previous glacial cycles based on the Antarctic methane and temperature records (with two different identification thresholds, red and blue circles in Fig. 9) following the approach of Siddall et al. (2006), found a high frequency of occurrence of Dansgaard-Oeschger events during MIS 3 and 6, but also throughout most other glacial phases. None of our simulations predicts such a ubiquity of 'excitable' AMOC modes, possibly due to the prescribed boundary conditions although the detection method of Barker et al. (2011) is also more uncertain for glacial cycles further back in time. The consistency of the simulated radiation-induced AMOC instability with observational indication of millennial-scale AMOC variability at least during MIS 3 and 6 in simulations A3 and A4 suggests that these could present a more realistic temporal AMOC evolution than the others. Simulations A3 and A4 also exhibit PI-LGM temperature differences of 5.4 and 6.2°C, respectively, close to the proxy-constrained reconstruction (Tierney et al., 2020), and roughly reproduce the reconstructed regional SST changes and reduced circulation strength in MIS 3 and 2 (Fig. 7 and 8).

Thermal conditioning of AMOC excitability is in line with studies that found the existence of a 'sweet spot' in atmospheric $CO_2$ radiative forcing which is particularly conducive to short, abrupt AMOC perturbations and/or self-sustained AMOC oscillations (e.g. Li and Born, 2019, Vettoretti et al., 2022). Yet, our simulations do not produce such perturbations, partly due to the smoothed forcing and static wind fields (see discussion of model limitations above). The transient circulation mode switches in response to orbitally-paced radiation changes in our simulations are much weaker than those found in other studies (Vettoretti et al., 2022, Klockmann et al., 2018, Kuniyoshi et al., 2022), and our simulations do not contain oscillations that could directly be compared to Dansgaard-Oeschger events.

## 4   Conclusions

Our study demonstrates the existence of thermal AMOC thresholds and multiple stable circulation modes in the Bern3D model. This adds to previous studies showing that thermal AMOC thresholds emerge in a range of Earth system models varying in complexity and number of components coupled (Zhang et al., 1993), in particular, they also arise in an energetically and hydrologically coupled ocean-sea ice-atmosphere model of intermediate

complexity like Bern3D. These thresholds shape the response in the simulated AMOC to
radiative orbital and atmospheric composition-driven temperature changes over the last 788
kyr. During this period the AMOC transitions between up to four persistent circulation modes.
The full glacial and interglacial circulation modes are most frequently simulated, as relatively
strong forcing is required to push the AMOC out of them. In contrast, the intermediate AMOC
modes are more sensitive to perturbations as small variations in orbital and radiative forcing
are able to push the circulation out of these modes. This behaviour resembles the one found
in more complex General Circulation Models that exhibit self-sustained oscillations at
'sweetspot' $CO_2$ levels, which lie between glacial and interglacial values. Our simulations
suggest that radiative forcing could have created time periods during which highly sensitive
intermediate AMOC modes occurred repeatedly over the last 788 kyr.

**Data availability**

All simulation output necessary to produce the figures in this manuscript are available at
https://doi.org/10.5281/zenodo.8424878
Proxy data plotted against the simulation output for comparison was taken from public
repositories and are available via the citations provided.

**Author contributions**

AJT ran the simulations. MA analysed the output and drafted the manuscript. All authors
contributed to the interpretation of the results and the final manuscript text.

**Conflicts of interest**

The authors declare that they have no conflict of interest.

**Acknowledgements**

MA, AJT and FJ were financially supported by the Swiss National Science Foundation
(#200020_200511).

FP was financially supported by the European Union's Horizon 2020 research and
innovation programme under grant agreements no. 101023443 (project CliMoTran).

TFS and FP were financially supported by the European Union's Horizon 2020 research and
innovation programme under grant agreements no. 820970 (project TiPES), and the Swiss
National Science Foundation's project 200020_200492.

Calculations were performed on UBELIX (http://www.id.unibe.ch/hpc), the HPC cluster at the
University of Bern.

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
