# Peer review of "Multiple thermal AMOC thresholds in the intermediate complexity model Bern3D"

_Climate of the Past, 2023_

## Referee Comment (RC2)

Review of manuscript by Adloff et al. "Multiple thermal AMOC thresholds in the intermediate complexity model Bern 3D"

In this manuscript, the authors performing several sets of transient experiments in Bern3D investigate thermally induced AMOC stability across glacial cycles. The results are new and complementary to our current theoretical understanding of glacial abrupt climate change. I believe this is a nice contribution to the community and suitable to Clim Past, but I reserve my recommendation for publication of this version since there remains room to improve its robustness and significance. In general, the authors shall 1) provide a more comprehensive introduction/discussion by considering at least most relevant literatures regarding AMOC stability during glacial cycles, 2) improve the clarity for mechanisms and feedback involved before, during and after AMOC transitions and 3) substantiate conclusions/statements by specifying the corresponding plots or adding direct modeling results/literatures. In addition, I would also recommend adding the 800-kyr results at least in the supplementary to provide an overview of the results, which would be of great interest for colleagues who are working on earlier glacial cycles as well.

Detailed comments are as follows:

P2L15-18: Freshwater input might be positive feedback to AMOC weakening as well. please refer to Barker et al 2015 and rephrase the sentences accordingly here as well as in L23-24.

P2L25-29: other key relevant paper should be cited, for instance, Zhang et al., 2014, 2017.

P2L33: also consider citing Zhang et al 2021; Vettoretti et al 2022 here.

P2L45-47: Please add relevant papers after the first sentence (e.g. Knorr and Lohmann 2007, Zhang et al 2017; Galbraith and de Lavergne 2018, etc.)

P4 L5: one predominant feature of glacial cycle is the development and demise of northern hemisphere ice sheet, involving both area and height, of which impacts on climate system are not the same. The former, as discussed in this study, via its albedo feedback is a thermal impact, while the latter, via its impacts on winds, is a kinetic impact (Zhang et al., 2014). In addition, there is no change in Bering Strait considered as well (Hu et al., 2011) (P5L4, a typo there). I was wondering how far these additional setups can alter the key messages of the thermal thresholds in this study. As seeing in my following comments, at least a comprehensive discussion around this is required.

P6 3.1: it would be good to present the 800kyr long transient simulation results. In Figure 1, it is of great help to add the radiative forcing curves to enable a comparison with B.slow experiment.

P7L15-17: As alluded, lacking feedback from topo changes might overestimate the LGM cooling caused by radiative forcing decrease because higher NHIS can cause a stronger AMOC which

promotes heat release from the ocean and hence North Atlantic warming. This might stimulate some discussion perhaps in data-model comparison or model limitation sections.

P8. Fig3: given the North Atlantic and Nordic Sea are the key regions for AMOC state shift, it would be better to provide a zoom-in plot for this region, especially for the sea ice fraction plot. Please also revise the color scheme for "sea ice cover fraction" to highlight change in the low values (<0.5) or just provide anomalous field as delta Density. Please also include lat-lon info in the plots. In addition, as you are discussing AMOC states, AMOC plots are highly recommended in this figure.

P8L18-20: in the state (II), deep water formation is enhanced in west and south of Greenland. In general, it is more reddish in State (II) than in State (I), but why the AMOC is weakened in the former. Is this due to that convection in the western North Atlantic is not the key to the strength of the AMOC?

P8L25: "south-flowing fresh Arctic waters further stratify …". This is a key process to stabilize the glacial AMOC state, but in this version, there is not direct evidence to support it. Note that freshwater convergence in Fig5e cannot provide such support to this statement because it is a sum of freshwater flux across both 40N and 70N in the North Atlantic.

P9L10: what is Kolmogrov-Smirnov test? Add details and reference.

P10 Fig 5: Panel e, it would be good to interpret meanings of positive/negative values of freshwater convergence to help readers understand this plot (e.g. positive values indicate freshwater import and hence a stable AMOC). In addition, the definition of freshwater convergence should be added to the Method section. It is worth noting that this AMOC stability indicator (Liu et al., 2014 Clim Dyn) predict a mono-stable AMOC regime in B.slow., in contrast to the hysteresis feature shown in Fig 4b. In addition, comparing the panel a) with Fig 4b, it appears that B.slow.b is initialized from a AMOC state that is bistable with respect to radiative forcing. If so, why the AMOC recovers to its initial strong mode after removing the freshwater input? Typo in y-axis labels of panel c). it is also good to add radiative forcing panel on the top of it, with a vertical shaded bar to highlight periods when AMOC is bistable.

P11L31: how do you identify the reduced heat convergence "off the British Isles" based on the time series in Fig5?

P11L33: It is also not logically clear why this is the cause to the northward spread of AABW. In Fig5, the northward intrusion of AABW is starting from the beginning of the experiment, not lagging the reduction of heat convergence in North Atlantic.

P11L35: why "heat advection to >55N stops entirely"? could the authors present the evidence?

P11L37-39: Again, no direct lines of evidence to support this statement. Does the contemporary sea ice expansion and its seasonality contribute to the freshening in the eastern Nordic Sea? As well as in P11L42-43. Please clarify.

Is there a bipolar thermal seesaw during abrupt AMOC reduction in B.slow? The results appear to show that bipolar sea ice change out of phase with AMOC/NADW change – sea ice expansion with NADW weakening. The subdued thermal seesaw in B.slow indicates the dominant role of decreasing radiative forcing in controlling bipolar change.

P12L27: what's the statement "… increased heat advection into the North Atlantic" based on?

P12L29: weakened north ward transport of what? Upper cell of the AMOC?

P13L1: please show the weakened the meridional salinity gradient in the North Atlantic.

P13L3-5: how does the increased surface density promote SST decrease? This is not clear at all here.

P13L5-7: the authors proposed that sea ice expansion over convection sites acts as negative feedback in response to SST cooling, which is not convincing. This process, as demonstrated in this sentence, can avoid further cooling of sea surface, which in turn reduced sea surface heat loss to increase surface density, and thus stratifying the water column. This seems to exert rather positive feedback to stabilizing the cooling-induced AMOC slowdown. Please clarify. In general, positive/negative feedback discussed in this paragraph is hard to follow. Please clarify with more direct evidence/references.

P13L11-13: As mentioned in previous comments, providing supportive evidence is of crucial importance since this is important positive feedback to the AMOC slow-down.

P13L15: please clarity and specify the positive and negative feedback mentioned here.

P13L22-23: given the gradual decreasing radiative forcing, it is not clear whether it is the self oscillation or just an increased variability (small magnitude, 0.5Sv) as the system approaches the threshold. It appears that AMOC variance is of comparable or even larger magnitude during 6-11kyr (Fig 5b). Is this also corresponding to self-oscillation?

P13P26-33: as discussed, results from B.slow.b seem not to support the hysteresis behavior with respect to radiative forcing change. What about stability/sensitivity of the AMOC at ~6kyr in B.slow?

P13L36: Orbital configuration consists of three orbital parameters. Their combinations in the chosen time slices are different but this does not mean the associated climatic impacts are significantly distinct, for instance, 21ka versus 0ka. It is thus better to show values of obliquity, precession, eccentricity and boreal summer insolation for the chosen time slices here, which

would be helpful to clarify whether orbital forcing matters the transient behavior of the AMOC. A better approach to test roles of orbital configurations is to re-conduct such transient experiments based on orbital sensitivity, for example, high versus low obliquity experiments (e.g. experiments in Extended Data Table 1 of Zhang et al 2021).

P14L5-7: not a full list of key relevant papers. Please add Knorr & Lohmann 2007, Banderas et al 2012 and Zhang et al 2017. Re multiple stable AMOC states, the difference in the strength of the AMOC is significantly different with a magnitude of >5Sv. In this context, it appears that the metastable AMOC states proposed here are perhaps sub-states of the interglacial/glacial AMOC state. Given the low AMOC variability in Bern3D, I assume this might not be reproducible by full GCMs nor perhaps in proxies.

P14L12-16: what's the exact role of 'heat advection" in AMOC mode transition? A positive feedback, a trigger or else? It would be good to have a clearer description here to specify the importance of heat advection.

P14L26-27: it is not true. For instance, Zhang et al 2017 applying a fully coupled AOGCM proposes that atmospheric CO2 levels are of control for glacial AMOC bi-stability.

P14L28-30: this may be true if comparing with other EMICs or simple models but not for GCM. Please clarify.

P14L35: please provide modeling results or relevant literatures to support this statement especially regarding poleward moisture transport. It appears to me Fig 5e would be the right panel to refer to given the different trends between Atlantic and North Atlantic freshwater convergence. Sentences in P11L13-14 seem already touch this point, but it requires future clarification to link them to moisture transport and so on.

P15: it is good to see the discussion about potential impacts of other parameters, especially ice sheet topography and associated wind, on the simulated AMOC change in different transient runs. In a glacial cycle, both changes in radiative forcing (e.g. CO2) and wind circulation/gateway caused by ice volume changes play a role in the strength/stability of the AMOC (Hu et al., 2011; Zhang et al 2014, 2017, 2021). Of most relevance here is their opposite impacts on the strength of the AMOC through glacial cycles in comparison to the thermal forcing (Barker and Knorr 2021). In this study, the authors investigated the roles of changes in radiative forcing in AMOC stability, which is the half story of AMOC multi-equilibria in glacial cycles. How do changes in those key parameters influence the results of A experiments? I would be happy to see more comprehensive discussion around this here as well as in Section 3.4 and 3.5. Perhaps, Section 3.3-3.5 can be integrated to one section to highlight and discuss the current understanding of AMOC stability, impacts of current model limitation on the current results and data-model comparison, and their implications and future perspectives.

P15 L14-15: Please add relevant reference to "different representations of processes affecting AABW density changes".

P17 Figure 9: it would be good to flip y-axis of d18O curve upside down, given the tradition of plotting LR04/sea level curves.

---

## Author Comment (AC1)

*We thank the reviewer for their time and effort reviewing our manuscript, and for the constructive comments which have helped to substantially improve our text. We address the reviewer's suggested improvements by completely rewriting our description and discussion of the processes that occur in the model, adding results to the figures in the main text and adding new figures to the SI, and referencing these figures more thoroughly in the text. Specifically, we make the changes outlined in our replies to the detailed comments below.*

*Below are our detailed point-by-point replies and suggested manuscript improvements (blue) for each comment (black).*

Detailed comments are as follows:

P2L15-18: Freshwater input might be positive feedback to AMOC weakening as well. please refer to Barker et al 2015 and rephrase the sentences accordingly here as well as in L23-24.

*We add the suggested reference and mention the possibility for freshwater feedbacks. Specifically, we add the following sentence to our introduction:*

*"Lags between the appearance of ice-rafted debris and the reconstructed cooling, however, suggest that freshwater fluxes could have instead acted as a positive feedback to AMOC weakening rather than triggering it (Barker et al., 2015)."*

P2L25-29: other key relevant paper should be cited, for instance, Zhang et al., 2014, 2017.

*We add references to Zhang et al., 2014 and 2017 and Vettoretti, 2022.*

P2L33: also consider citing Zhang et al 2021, Vettoretti et al 2022 here.

*We add the suggested references.*

P2L45-47: Please add relevant papers after the first sentence (e.g. Knorr and Lohmann 2007, Zhang et al 2017, Galbraith and de Lavergne 2018, etc.)

*We add the suggested references.*

P4 L5: one predominant feature of glacial cycle is the development and demise of northern hemisphere ice sheet, involving both area and height, of which impacts on climate system are not the same. The former, as discussed in this study, via its albedo feedback is a thermal impact, while the latter, via its impacts on winds, is a kinetic impact (Zhang et al., 2014). In addition, there is no change in Bering Strait considered as well (Hu et al., 2011) (P5L4, a typo there). I was wondering how far these additional setups can alter the key messages of the thermal thresholds in this study. As seeing in my following comments, at least a comprehensive discussion around this is required.

*We add more discussion of other factors for AMOC stability, see also our answers to further comments on this topic below. We will add the following text to section 3.3:*

*"Previous studies suggested that pre-industrial or intermediate glacial ice sheet configurations are required to even produce a thermal AMOC threshold in the range of glacial-interglacial $CO_2$ concentrations in a full GCM and that the presence of a full glacial Laurentide ice sheet prevents*

such a threshold (e.g. Klockmann et al., 2018). In addition, changes in the interconnection of marine basins, specifically the Bering Strait, also affects AMOC stability (Hu et al. 2012). The values of the thermal thresholds in our experiments are thus likely sensitive to the model design and initiation. Pöppelmeier et al. (2021) showed that the sensitivity of Bern3D to freshwater hosing increases when additional LGM boundary conditions are prescribed (changed wind fields, closed Bering Strait, tidal mixing differences due to sea level changes). The different wind fields and tidal mixing strengthened AMOC and increased the salt and heat transport into the subpolar North Atlantic. This could mean that stronger cooling is required to stabilise the water column in the Irminger Sea and reach the first thermal threshold, when the full range of glacial boundary conditions are applied. Closure of the Bering Strait increased the salt advection feedback, which stabilises the weak circulation state without deepwater formation in the subpolar North Atlantic. Further investigations are needed to determine how changes in strength and location of the wind stress due to the ice sheet's orography, sea level and Bering strait closure would affect sea ice formation in the northern North Atlantic and the AMOC thresholds in our simulations quantitatively. "

P6 3.1: it would be good to present the 800kyr long transient simulation results. In Figure 1, it is of great help to add the radiative forcing curves to enable a comparison with B.slow experiment.

*We add the according figure to the SI.*

P7L15-17: As alluded, lacking feedback from topo changes might overestimate the LGM cooling caused by radiative forcing decrease because higher NHIS can cause a stronger AMOC which promotes heat release from the ocean and hence North Atlantic warming. This might stimulate some discussion perhaps in data-model comparison or model limitation sections.

*We add this to our discussion as suggested (see the previously shown addition to section 3.3.)*

P8. Fig3: given the North Atlantic and Nordic Sea are the key regions for AMOC state shift, it would be better to provide a zoom-in plot for this region, especially for the sea ice fraction plot. Please also revise the color scheme for "sea ice cover fraction" to highlight change in the low values (<0.5) or just provide anomalous field as delta Density. Please also include lat-lon info in the plots. In addition, as you are discussing AMOC states, AMOC plots are highly recommended in this figure.

*We change the figure (or panel?) to show sea ice anomalies in the North Atlantic and add coordinates. We also add vector plots showing AMOC circulation to the SI.*

P8L18-20: in the state (II), deep water formation is enhanced in west and south of Greenland. In general, it is more reddish in State (II) than in State (I), but why the AMOC is weakened in the former. Is this due to that convection in the western North Atlantic is not the key to the strength of the AMOC?

*Yes, the mixed layer depth that we diagnosed from the annually-averaged model output is not a good metric to understand changes in AMOC strength. We'll explain the density changes more explicitly in the new manuscript version, and include that some changes in the locations of downwelling occur without changing AMOC strength. We'll remove the plots of mixed layer depth and instead show the absolute vertical density gradients in each state. We will add the following text to section 3.1:*

*"Initially, the whole Atlantic surface ocean cools and freshens, leaving the temperature and salinity differences between the Irminger and Caribbean Seas almost unchanged (Fig 5e). However, NADW becomes less salty and colder as a consequence of the changes in the surface ocean (not shown) and the vertical density profiles in the subpolar North Atlantic steepen due to the temperature and salinity changes (Fig. SI.7-8). After about 6 kyr, the changes in the North Atlantic density profile shift the location of NADW formation. NADW formation moves south as vertical density profiles in the subpolar east North Atlantic stabilise under a freshening of the surface and density profiles further south steepen due to surface cooling combined with subsurface warming (Fig. SI.7-9). These changes do not cause a step-change in AMOC strength, but freshwater and heat advection into the North Atlantic is reduced, sea ice expansion increases in the eastern North Atlantic and AMOC variance (calculated over a moving 50-year window) is increased (Fig. 5). Transport of heat and salinity into the North Atlantic decreases (Fig. 5f, g) and North Atlantic SST and SSS decrease (Fig. 5e). Reduced influx of subtropical surface waters also causes sudden cooling and freshening in the Irminger Sea (Fig. SI.8)."*

P8L25: "south-flowing fresh Arctic waters further stratify …". This is a key process to stabilize the glacial AMOC state, but in this version, there is not direct evidence to support it. Note that freshwater convergence in Fig5e cannot provide such support to this statement because it is a sum of freshwater flux across both 40N and 70N in the North Atlantic.

*We show the freshwater fluxes across each latitude separately in the updated Fig. 5. We also clarify that the spread of cold, fresh surface water in the North Atlantic stabilises the circulation state. We cannot actually determine whether the water comes from the Arctic or just turns more Artic-like due to reduced influx of southern surface water.*

P9L10: what is Kolmogrov-Smirnov test? Add details and reference.

*We provide some more information and a reference:*

*"but none are statistically significant in the two-sided Smirnov test, which determines the likelihood that two distributions are the same (Berger and Zhou, 2014), even at the 50% confidence level"*

P10 Fig 5: Panel e, it would be good to interpret meanings of positive/negative values of freshwater convergence to help readers understand this plot (e.g. positive values indicate freshwater import and hence a stable AMOC). In addition, the definition of freshwater convergence should be added to the Method section. It is worth noting that this AMOC stability indicator (Liu et al., 2014 Clim Dyn) predict a mono-stable AMOC regime in B.slow., in contrast to the hysteresis feature shown in Fig 4b. In addition, comparing the panel a) with Fig 4b, it appears that B.slow.b is initialized from a AMOC state that is bistable with respect to radiative forcing. If so, why the AMOC recovers to its initial strong mode after removing the freshwater input? Typo in y-axis labels of panel c). it is also good to add radiative forcing panel on the top of it, with a vertical shaded bar to highlight periods when AMOC is bistable.

*We amend Figure 5 as suggested. The hysteresis shown in Fig 4b is not the result of a traditional perturbation experiment with freshwater hosing but is the transient response to the applied radiative forcing. We are not sure if stability with regard to a freshwater perturbation is the same as stability in the face of changing boundary conditions, as caused by the radiative forcing. We are therefore careful with the interpretation of freshwater transport as stability indicators in our study. We add the following text to section 3.1:*

*"The difference between freshwater transport into the South Atlantic at 32°S and into the Arctic at 62.5°N in Fig. 5f can be used as a measure for the basin-wide salinity feedback (Rahmstorf, 1996, de Vries and Weber, 2005). In our simulation, changes in this metric are predominantly caused by changes in the transport across the northern edge, since transport into the South Atlantic remains almost unchanged throughout the cooling phase of B.slow. North Atlantic salinity is instead governed by changing transport from the subtropics into the North Atlantic and between the North Atlantic and Arctic. As such, the processes involved in the sudden AMOC strength changes, namely density changes in the upper water column, and those that stabilise new circulation states (salinity and heat redistributions, sea ice expansion) mostly operate in the North Atlantic region."*

P11L31: how do you identify the reduced heat convergence "off the British Isles" based on the time series in Fig5?

*The geographic information is not derived from the time series but provided as extra information to contextualise the time series. We add a figure to SI showing the discussed spatial pattern (new Fig. SI.11).*

P11L33: It is also not logically clear why this is the cause to the northward spread of AABW. In Fig5, the northward intrusion of AABW is starting from the beginning of the experiment, not lagging the reduction of heat convergence in North Atlantic.

*We agree, and change the sentence to point out coincidence rather than causality. The experiment is initialised with no AABW tracers in the North Atlantic. Initially, changes in the concentration of AABW tracers in the North Atlantic are small. Their amount only begins to rise substantially at ~15 kyr and shows the biggest jump at ~27 kyr when the heat convergence also declines.*

P11L35: why "heat advection to >55N stops entirely"? could the authors present the evidence?

*We apologise, this should have been 'heat convergence'. We correct this and add a figure of the spatial pattern of heat convergence changes to the SI (new Fig. SI.11).*

P11L37-39: Again, no direct lines of evidence to support this statement. Does the contemporary sea ice expansion and its seasonality contribute to the freshening in the eastern Nordic Sea? As well as in P11L42-43. Please clarify. Is there a bipolar thermal seesaw during abrupt AMOC reduction in B.slow? The results appear to show that bipolar sea ice change out of phase with AMOC/NADW change – sea ice expansion with NADW weakening. The subdued thermal seesaw in B.slow indicates the dominant role of decreasing radiative forcing in controlling bipolar change.

*Yes, we thank the reviewer for pointing this out. There is a small bipolar seesaw effect. We mention this explicitly in the revised manuscript. We add a plot of with spatial patterns of changes in B.slow to the SI and add the following text to section 3.1:*

*"The biggest AMOC weakening at ~27 kyr is also accompanied by a weak bipolar seesaw effect, which causes a temporary decline in sea ice coverage in the Atlantic sector of the Southern Ocean (Fig. 5). It is, however, too small to reduce the radiation-driven sea ice increase in the longer term."*

P12L27: what's the statement "… increased heat advection into the North Atlantic" based on?

*This statement was erroneous and we delete it. AMOC strength is not constant at the beginning of the experiment but weakens slowly, while the spatial pattern of deep convection and heat*

*convergence in the North Atlantic change. We rewrite our description of the processes responsible for AMOC changes and use more references to figures. We add the spatial patterns of heat convergence changes to the new Fig. SI. SI.*

P12L29: weakened north ward transport of what? Upper cell of the AMOC?

*We clarify that we speak of transport of salt and heat.*

P13L1: please show the weakened the meridional salinity gradient in the North Atlantic.

*We apologise, this was meant to say weakened meridional salinity transport, i.e. an increased salinity gradient. We correct the statement in the text and add the temporal evolution of salinity in the Irminger and Caribbean Seas to Fig. 5 to show the increased meridional salinity gradient.*

P13L3-5: how does the increased surface density promote SST decrease? This is not clear at all here.

*We agree with the reviewer that this formulation was misleading. We meant to express that SST changes have a direct effect on water density and an indirect one via influencing evaporation, and that temperature-driven evaporation changes counteract the buoyancy forcing caused by the temperature change. We will rewrite this paragraph as follows:*

*"The deep water formation regions are sensitive to heat and salt flux changes, because any reduction in sea surface temperatures (SST) increases surface density but simultaneously reduces evaporation in ice-free areas, thus effectively creating a small freshwater forcing and a negative feedback to the buoyancy changes caused by the initial SST decrease. Sea ice covering the downwelling areas stabilise the water column by preventing surface ocean cooling and evaporation. The progressive influx of AABW into the North Atlantic is a further process stabilising new circulation states by stratifying the water column from below (Buizert and Schmittner, 2015)."*

P13L5-7: the authors proposed that sea ice expansion over convection sites acts as negative feedback in response to SST cooling, which is not convincing. This process, as demonstrated in this sentence, can avoid further cooling of sea surface, which in turn reduced sea surface heat loss to increase surface density, and thus stratifying the water column. This seems to exert rather positive feedback to stabilizing the cooling-induced AMOC slowdown. Please clarify. In general, positive/negative feedback discussed in this paragraph is hard to follow. Please clarify with more direct evidence/references.

*We agree with the reviewer that the discussion of feedbacks in this paragraph is unclear and we will rewrite this. Importantly, the current version suggests that sea ice cover is a negative feedback on SST changes, which is not correct. Sea ice cover prevents evaporation and heat loss to the atmosphere, stabilising the water column. This would be better described as removing the positive feedback buoyancy changes. We will rewrite the paragraph focussed on stabilising mechanisms rather than feedbacks as follows:*

*"The resulting redistributions of heat and salinity cause sudden shifts in the vertical density profiles and sea ice expansion which consolidate the new circulation state. The downwelling zones are sensitive to heat and salt flux changes, because any reduction in sea surface temperatures (SST) increases surface density but simultaneously reduces evaporation in ice-free areas, thus effectively creating a small freshwater forcing and a negative feedback to the buoyancy changes caused by*

the initial SST decrease. Sea ice covering the downwelling areas stabilise the water column by preventing surface ocean cooling and evaporation. The progressive influx of AABW into the North Atlantic is a further process stabilising new circulation states by stratifying the water column from below (Buizert and Schmittner, 2015).. The difference between freshwater transport into the South Atlantic at 32°S and into the Arctic at 62.5°N in Fig. 5f can be used as a measure for the basin-wide salinity feedback (Rahmstorf, 1996, de Vries and Weber, 2005). In our simulation, changes in this metric are predominantly caused by changes in the transport across the northern edge, since transport into the South Atlantic remains almost unchanged throughout the cooling phase of B.slow. North Atlantic salinity is instead governed by changing transport from the subtropics into the North Atlantic and between the North Atlantic and Arctic. As such, the processes involved in the sudden AMOC strength changes, namely density changes in the upper water column, and those that stabilise new circulation states (salinity and heat redistributions, sea ice expansion) mostly operate in the North Atlantic region."

P13L11-13: As mentioned in previous comments, providing supportive evidence is of crucial importance since this is important positive feedback to the AMOC slow-down.

*We make more references to figures in the re-written paragraph.*

P13L15: please clarity and specify the positive and negative feedback mentioned here.

*We agree that this paragraph is unclear, and we will rewrite it by using the clearer term 'stabilising process' (see above).*

P13L22-23: given the gradual decreasing radiative forcing, it is not clear whether it is the self oscillation or just an increased variability (small magnitude, 0.5Sv) as the system approaches the threshold. It appears that AMOC variance is of comparable or even larger magnitude during 6-11kyr (Fig 5b). Is this also corresponding to self-oscillation?

*We are now more cautious with our statement and only write that variability is increased. The large variability at 6-11 kyr is related to density changes in the Irminger Sea. We discuss this in the new manuscript version in section 3.1, but we didn't see indications of oscillations:*

"After about 6 kyr, the changes in the North Atlantic density profile result in shifts in the spatial pattern of NADW formation. NADW formation moves south as vertical density profiles in the subpolar east North Atlantic stabilise under a freshening of the surface and density profiles further south steepen due to surface cooling combined with subsurface warming (Fig. SI.7-9). Apart from temporary volatility, the mean AMOC strength is not affected by these changes, but freshwater and heat advection into the North Atlantic is reduced, sea ice expansion increases in the eastern North Atlantic and AMOC variance (calculated over a moving 50-year window) is increased (Fig. 5)."

P13P26-33: as discussed, results from B.slow.b seem not to support the hysteresis behavior with respect to radiative forcing change. What about stability/sensitivity of the AMOC at ~6kyr in B.slow?

*We referred to hysteresis behaviour here because the radiative forcing that is required to cause the abrupt weakening of AMOC is not the same as the forcing required for strengthening it again. We are clearer in the revised manuscript. We did not test AMOC stability explicitly at 6 kyr because the high variability seems to cease once the density field has re-adjusted.*

P13L36: Orbital configuration consists of three orbital parameters. Their combinations in the chosen time slices are different but this does not mean the associated climatic impacts are significantly distinct, for instance, 21ka versus 0ka. It is thus better to show values of obliquity, precession, eccentricity and boreal summer insolation for the chosen time slices here, which would be helpful to clarify whether orbital forcing matters the transient behavior of the AMOC. A better approach to test roles of orbital configurations is to re-conduct such transient experiments based on orbital sensitivity, for example, high versus low obliquity experiments (e.g. experiments in Extended Data Table 1 of Zhang et al 2021).

*We provide the orbital parameter values for each experiment in the SI. We were mostly interested here to see that changing the orbital configuration does not substantially alter the simulation results. It would be interesting to investigate the role of orbital changes for thermal thresholds in more detail in the future.*

P14L5-7: not a full list of key relevant papers. Please add Knorr & Lohmann 2007, Banderas et al 2012 and Zhang et al 2017. Re multiple stable AMOC states, the difference in the strength of the AMOC is significantly different with a magnitude of >5Sv. In this context, it appears that the metastable AMOC states proposed here are perhaps sub-states of the interglacial/glacial AMOC state. Given the low AMOC variability in Bern3D, I assume this might not be reproducible by full GCMs nor perhaps in proxies.

*We thank the reviewer for these additional relevant references, which we add to the paragraph. We also add further SI figures that show that each of the four persistent AMOC strengths is associated with different Greenland temperatures and North Atlantic sea ice extents, suggesting that they correspond to different climate states. Further, we agree, that it would be interesting to see this tested with a full GCM in future studies*

P14L12-16: what's the exact role of 'heat advection" in AMOC mode transition? A positive feedback, a trigger or else? It would be good to have a clearer description here to specify the importance of heat advection.

*Yes, we clarify this. Changes in heat convergence only seem to stabilise density profiles in the North Atlantic rather than causing AMOC shifts. We add these two paragraphs to section 3.1:*

*"After about 6 kyr, the changes in the North Atlantic density profile result in shifts in the spatial pattern of NADW formation. NADW formation moves south as vertical density profiles in the subpolar east North Atlantic stabilise under a freshening of the surface and density profiles further south steepen due to surface cooling combined with subsurface warming (Fig. SI.7-9). Apart from temporary volatility, the mean AMOC strength is not affected by these changes, but freshwater and heat advection into the North Atlantic is reduced, sea ice expansion increases in the eastern North Atlantic and AMOC variance (calculated over a moving 50-year window) is increased (Fig. 5). Transport of heat and salinity into the North Atlantic decreases (Fig. 5f, g) and North Atlantic SST and SSS decrease (Fig. 5e). Reduced influx of Atlantic waters also causes sudden cooling and freshening in the Irminger Sea (Fig. SI.8)."*

*"The simulated step changes in AMOC strength in our simulations are thus the response to gradual surface cooling and freshening, and occur when NADW formation shifts southwards. The resulting redistribution of heat and salinity cause sudden shifts in the vertical density profiles and sea ice expansion which consolidate the new circulation state."*

P14L26-27: it is not true. For instance, Zhang et al 2017 applying a fully coupled AOGCM proposes that atmospheric CO2 levels are of control for glacial AMOC bi-stability.

*Our intention here was to understand why Oka et al. (2021), specifically, required a stronger forcing in the southern hemisphere for thermal thresholds to arise, while we see thermal thresholds in our model under a globally uniform forcing. We clarify this and also refer to Zhang et al. (2017):*

"In their examination of thermal forcing of both hemispheres in an ocean-only model, Oka et al. (2021) found that thermal AMOC thresholds only exist in the ocean model COCO if the Southern Hemisphere is cooled more than the Northern Hemisphere. In contrast, Zhang et al. (2017) found sudden AMOC changes also due to changes in well-mixed greenhouse gases. In our simulations with Bern3D, we also find thermal thresholds with similar cooling rates in both hemispheres, but only after salinity redistributions and changing meteoric freshwater fluxes in response to about six thousand years of global cooling. In accordance with Lohmann et al. (2023), we found that the observed shifts in AMOC strengths were thus preceded by cascades of density and circulation field changes, the number and sequence of which depend on the strength of the forcing. It would be interesting to test whether models seemingly without thermal thresholds under North Hemispheric cooling can reach such thresholds on the timescale of tens of thousand years. It is also possible that changing meteoric freshwater fluxes are essential for the existence of such a thermal threshold, which does not therefore appear in an ocean model without a thermally responsive atmosphere with a climate-driven freshwater balance."

P14L28-30: this may be true if comparing with other EMICs or simple models but not for GCM. Please clarify.

*This was a wrong conception, we remove this statement from the manuscript.*

P14L35: please provide modeling results or relevant literatures to support this statement especially regarding poleward moisture transport. It appears to me Fig 5e would be the right panel to refer to given the different trends between Atlantic and North Atlantic freshwater convergence. Sentences in P11L13-14 seem already touch this point, but it requires future clarification to link them to moisture transport and so on.

*The statement on moisture transport was specific to our model. The wind field is constant but the water holding capacity of air decreases with the temperature decline. Hence, less moisture is transported polewards by the large-scale atmospheric circulation. We clarify this in the text. We also add more specific metrics of the changing water balance to the figures. We add SSS and SST timeseries for the Caribbean and Irminger Sea, as well as marine freshwater fluxes across latitudes 37.5°N and 62.5°N to Fig. 5, and spatial changes of P-E in the North Atlantic to the SI.*

*The additional text in section 3.3 is:*

"In a model with a climate-sensitive meteoric freshwater balance, climate cooling reduces evaporation and the water-holding capacity. In such a model, cooling can then affect seawater density directly via changing temperatures, and indirectly via changing the meteoric freshwater balance and seawater salinity. In this context, it is also important to consider spatial changes in atmospheric dynamics, which are kept constant in our simulations, i.e. the moisture content of air changes with climate but not the direction or strength of winds which disperse it. In our model, a decrease in the water-holding capacity of air therefore directly leads to a reduction of the large scale atmospheric moisture transport from low to high latitudes. Several studies found that wind

stress changes in the North Atlantic and Southern ocean are important controls on AMOC stability (e.g. Arzel et al., 2008, Yang et al., 2016) and thermal thresholds (Oka et al., 2012). The specific effects of atmospheric dynamics on meteoric freshwater forcing on AMOC would be an additional relevant topic for future studies."

P15: it is good to see the discussion about potential impacts of other parameters, especially ice sheet topography and associated wind, on the simulated AMOC change in different transient runs. In a glacial cycle, both changes in radiative forcing (e.g. CO2) and wind circulation/gateway caused by ice volume changes play a role in the strength/stability of the AMOC (Hu et al., 2011, Zhang et al 2014, 2017, 2021). Of most relevance here is their opposite impacts on the strength of the AMOC through glacial cycles in comparison to the thermal forcing (Barker and Knorr 2021). In this study, the authors investigated the roles of changes in radiative forcing in AMOC stability, which is the half story of AMOC multi-equilibria in glacial cycles. How do changes in those key parameters influence the results of A experiments? I would be happy to see more comprehensive discussion around this here as well as in Section 3.4 and 3.5. Perhaps, Section 3.3-3.5 can be integrated to one section to highlight and discuss the current understanding of AMOC stability, impacts of current model limitation on the current results and data-model comparison, and their implications and future perspectives.

*We follow the reviewer's advice and add more discussion of AMOC stability and model limitations by combining sections 3.3 and 3.5. The new paragraph reads:*

"In our transient simulations covering the past 788 kyr, the AMOC strength decreases during glacial phases solely due to changes in the hydrological cycle and sea ice that are induced by orbital, greenhouse gas and dust-driven temperature changes. The existence of multiple stable AMOC modes under varying thermal or radiative forcings has been found in various GCMs (e.g. Knorr and Lohmann, 2007, Oka et al., 2012, Banderas et al., 2012, Brown and Galbraith, 2016, Zhang et al., 2017, Klockmann et al., 2018). In agreement with previous studies, we found multiple (meta)stable AMOC circulation modes with distinct AMOC strengths for radiative forcing levels between full glacial and interglacial climate states. Moreover, we find that the transitions between these states occur abruptly, some within as little as 100 years. In accordance with Lohmann et al. (2023), we found that these shifts in AMOC strengths were thus preceded by cascades of density and circulation field changes, the number and sequence of which depend on the strength of the forcing. Similar to the findings from Oka et al. (2021), these AMOC transitions arise primarily from salt redistribution in the ocean and sea ice expansion into deep convection zones. In our simulations, each transition in AMOC strength is associated with a shift in the convergence of heat and salt fluxes and a southward expansion of sea ice into the North Atlantic which increasingly decouples the surface ocean buoyancy from the atmosphere. In the meta-stable modes, the density gradients in the main North Atlantic deep convection zones are strongly dependent on surface buoyancy fluxes. In these modes, small changes in buoyancy or sea ice cover can cause resumption or cessation of convection, which makes the AMOC sensitive to small perturbations. The AMOC is only pushed into its weakest mode when the net heat advection into the North Atlantic is strongly reduced and all former convection sites in the subpolar North Atlantic are sea ice-covered. In their examination of thermal forcing of both hemispheres in an ocean-only model, Oka et al. (2021) found that thermal AMOC thresholds only exist in the ocean model COCO if the Southern Hemisphere is cooled more than the Northern Hemisphere. In contrast, Zhang et al. (2017) found sudden AMOC changes also due to greenhouse gas changes without a special focus on the Southern Hemisphere. In our simulations with Bern3D, we also found thermal thresholds with similar cooling rates in both hemispheres, but only after salinity redistributions and changing meteoric freshwater fluxes in response to about six thousand years of global cooling.

It is possible that changing meteoric freshwater fluxes are essential for the existence of such a thermal threshold, which does not therefore appear in an ocean model without a thermally responsive atmosphere with a climate-driven freshwater balance. In a model with a dynamic energy moisture balance component, atmospheric cooling reduces evaporation, the water-holding capacity of the atmosphere and the atmospheric poleward transport of moisture. In such a model, cooling can then affect seawater density directly via changing temperatures, and indirectly via changing the meteoric freshwater balance and surface salinities. These changes would induce additional kinematic changes (i.e., in the wind fields) in fully dynamic atmosphere models, but are kept constant in our simulations, i.e. the moisture content of air changes with climate but not the direction or strength of winds which disperse it. Accordingly, wind stress fields are also kept constant here. Changes in wind stress have been documented to exert important controls on AMOC stability (e.g. Arzel et al., 2008, Yang et al., 2016) and thermal thresholds (Oka et al., 2012). These effects have been investigated in detail with the Bern3D model by Pöppelmeier et al. (2021) focusing on LGM boundary conditions.

The primary importance of salinity and heat redistributions as well as sea ice extent in the North Atlantic for the simulated AMOC shifts resembles the findings from Ando and Oka (2021)'s hosing experiments under LGM conditions and Zhang et al. (2017)'s simulations of AMOC shifts in response to CO2 changes in intermediate-glacial conditions. While our experiments are run with pre-industrial boundary conditions, the initial location of convection sites between Greenland and the British Islands (areas with lowest density differences over upper 1000m in Fig. SI.8) resembles the LGM and intermediate-glacial circulation states in Ando and Oka (2021) and Zhang et al. (2017). Ganopolski and Rahmstorf (2001) found that the possibility of a southward shift of deep convection depends on the latitude of prior deep convection and the density field further south, and Oka et al. (2012) showed that the location of deep convection and its distance from the winter sea ice edge define thermal thresholds in AMOC strength. Several controls on the location and strength of deep convection in the North Atlantic, that would have affected AMOC stability over glacial cycles, have been established. The location of deep convection is dependent on wind fields, climate and sea level/bathymetry (Ganopolski and Rahmstorf, 2001, Oka et al., 2012, Zhang et al., 2017), and thus the thermal AMOC thresholds are model and forcing dependent (Oka et al., 2012). Our simulations capture the albedo effect of varying terrestrial ice sheet extent, but we do not consider their orography or sea level effects, including impacts on the atmospheric circulation, which were shown to affect AMOC (Li and Born, 2019, Pöppelmeier et al., 2021). Previous studies suggested that pre-industrial or intermediate ice sheet configurations are required to even produce a thermal AMOC threshold in the range of glacial-interglacial $CO_2$ concentrations and that the presence of a full glacial Laurentide ice sheet prevents such a threshold (e.g. Klockmann et al., 2018). In addition, changes in the interconnection of marine basins, specifically the Bering Strait, also affect AMOC stability (Hu et al. 2012). The values of the thermal thresholds in our experiments are thus likely sensitive to the model design and initiation. Pöppelmeier et al. (2021) showed that the sensitivity of Bern3D to freshwater hosing increases when additional LGM boundary conditions are prescribed (changed wind fields, closed Bering Strait, tidal mixing differences due to sea level changes). The different wind fields and tidal mixing strengthened AMOC and increased the salt and heat transport into the subpolar North Atlantic. This could mean that stronger cooling is required to stabilise the water column in the Irminger Sea and reach the first thermal threshold, when the full range of glacial boundary conditions are applied. Closure of the Bering Strait increased the salt advection feedback, which stabilises the weak circulation state without deep water formation in the subpolar North Atlantic.

Further investigations are needed to determine how changes in strength and location of the wind stress due to the ice sheet's orography, sea level and Bering strait closure would affect sea ice formation in the northern North Atlantic and the AMOC thresholds in our simulations quantitatively. Since we chose to focus on radiation driven AMOC changes in our experiments, we would not expect a close model-data match with reconstructed AMOC changes from paleo-records. Our simulations show that the reconstructed temperature changes had the potential to alter the density field in the North Atlantic by redistributing heat and salt, and that some of these changes might have resulted in abrupt changes of AMOC strength. By testing a wide range of glacial-interglacial temperature changes, our experiments demonstrate that the cooling during glacial periods likely contributed to a weakened AMOC. The strength and timing of the weakening depends on the actual temperature change in the North Atlantic which would have been modulated by changes in winds and ice shields."

P15 L14-15: Please add relevant reference to "different representations of processes affecting AABW density changes".

*We add a reference here:* "(e.g. brine rejection, Bouttes et al., 2011)"

P17 Figure 9: it would be good to flip y-axis of d18O curve upside down, given the tradition of plotting LR04/sea level curves.

*We invert the y-axis of the $\delta^{18}O$ panel as suggested.*

References

[revised manuscript text omitted]

---

## Author Comment (AC2)

*We thank the reviewer for the invested time in evaluating our study and the thoughtful comments that have helped to substantially improve the manuscript.*
*Below are our detailed point-by-point replies and suggested manuscript improvements (blue) for each comment (black).*

Main comments:

Mechanisms by which the reduction of radiative forcing weakens the AMOC: Other reviewers have pointed out many points, so I'll just list potential ways to improve the manuscript.

1. Separate the paragraph explaining the effect of the Southern Ocean and North Atlantic. Section 3.2 goes back and forth between the role of NA and Southern Ocean. This makes it hard to follow the discussion. Related to this, Buizert and Schmittner (2015) provides a nice summary on the role of Southern Ocean. Ando and Oka (2021, GRL) also gives useful insight on the role of sea ice and heat transport on the stability of the AMOC. These two studies further performed hysteresis experiments with freshwater forcing. While the way of hysteresis experiment is not the same as in this study, I feel these studies should be cited and included in the discussion of the mechanism.

*We rewrite the description of the processes at play in simulation B.slow, and follow the reviewer's advice to discuss changes in the North Atlantic and Southern Ocean separately. We will also refer to Buizert and Schmittner (2015) and Ando and Oka (2021) in the discussion of our results. The revised paragraphs read as follows:*

*In section 3.2:*
*"In our simulations, the primary processes controlling the AMOC strength under changing radiative forcing are density changes due to heat and salinity redistributions. We investigate these in more detail in experiment B.slow (Fig. 4 and 5). This experiment is characterised by a slow linear decrease in radiative forcing over 50 kyr, before it is increased again to the pre-industrial value with the same rate of change(Fig. 4a). Fig. 5 shows that AMOC weakens gradually over the first 24 kyr, then weakens abruptly by 1 Sv at 24 kyr into the simulation and by ~3 Sv at 27 kyr, and then continues to weaken gradually until the forcing is reversed (Fig. 5a). In addition to the abrupt transition in AMOC strength, we found several additional rapid changes in AMOC variability, heat and salt fluxes (Fig. 5) and regional density profiles (Fig. SI.7-9), which are not associated with abrupt changes in AMOC strength. In fact, experiment B.slow shows that a cascade of changes with little effect on the mean AMOC strength occur before the first abrupt AMOC weakening after 24 kyr. Since these changes might partially be artefacts of our coarse model resolution, we here only focus on the larger scale changes instead. Initially, the whole Atlantic surface ocean cools and freshens, leaving the temperature and salinity differences between the Irminger and Caribbean Seas almost unchanged (Fig 5e). However, NADW becomes less salty and colder in consequence (not shown) and the vertical density profiles in the subpolar North Atlantic change due to the temperature and salinity changes (Fig. SI.7-8).*
*After about 6 kyr, the changes in the North Atlantic density profile shift the location of NADW formation. NADW formation moves south as vertical density profiles in the subpolar east North Atlantic stabilise under a freshening of the surface and density profiles further south steepen due to surface cooling combined with subsurface warming (Fig. SI.7-9). Apart from temporary volatility, the mean AMOC strength is not affected by these changes, but freshwater and heat advection into the North Atlantic is reduced, sea ice expansion increases in the eastern North Atlantic and AMOC variance (calculated over a moving 50-year window) is increased (Fig. 5). Transport of heat and salinity into the North Atlantic decreases (Fig. 5f, g) and North Atlantic SST and SSS decrease (Fig. 5e). Reduced influx of subtropical surface waters also cause abrupt cooling and freshening in*

the Irminger Sea (Fig. SI.8). At 24 kyr, the AMOC has weakened to ~14.5 Sv and sea ice cover extends south of the Irminger Sea (Fig SI.10). At this point, the AMOC strength drops abruptly by 1 Sv, and then by an additional 2.5 Sv ~3 kyr later, as the reduced salinity advection into the North Atlantic and precipitation and evaporation changes lead to a strong surface freshening. As a result of the North Atlantic density changes, the main North Atlantic convection site shifts southwards (determined by changes in the vertical density profiles, Fig SI.10). Sea ice also increasingly covers former areas of deep water formation in the North Atlantic. In the weakest circulation mode, the location of the maximum AMOC streamfunction shifts southwards by approximately 10 degrees and up in the water column by 400 m initially (28.5 kyr) and eventually almost 800 m (47 kyr) This shift allows cold, less dense water masses to extend further south into the North Atlantic.

In the Southern Ocean, the cooling enhances Southern Ocean deep water formation early on in the experiment and leads to a continuous expansion of sea ice in the Southern Hemisphere. The biggest AMOC weakening at ~27 kyr is also accompanied by a weak bipolar seesaw effect, which causes a temporary decline in sea ice coverage in the Atlantic sector of the Southern Ocean (Fig. 5). It is, however, too small to reduce the radiation-driven sea ice increase in the longer term. Both shifts in AMOC strength are accompanied by an increased spread of AABW into the North Atlantic (Fig. 5d)."

"The simulated step changes in AMOC strength in our simulations are thus the response to gradual surface cooling and freshening, and occur when NADW formation shifts southwards. The resulting redistributions of heat and salinity cause sudden shifts in the vertical density profiles and sea ice expansion which consolidate the new circulation state (Ando and Oka, 2021). In particular, reduced advection of heat and salinity into former locations of deep water formation result in a more stable local water column (Fig. SI.7-9). The downwelling zones are sensitive to heat and salt flux changes, because any reduction in sea surface temperatures (SST) increases surface density but simultaneously reduces evaporation in ice-free areas, thus effectively creating a small freshwater forcing and a negative feedback to the buoyancy changes caused by the initial SST decrease. Sea ice covering the downwelling areas stabilises the water column by preventing surface ocean cooling and evaporation. The progressive influx of AABW into the North Atlantic is a further process stabilising new circulation states by stratifying the water column from below (Buizert and Schmittner, 2015)."

*In section 3.3:*
"The primary importance of salinity and heat redistributions as well as sea ice extent in the North Atlantic for the simulated AMOC shifts resembles the findings from Ando and Oka (2021)'s hosing experiments under LGM conditions and Zhang et al. (2017)'s simulations of AMOC shifts in response to CO2 changes under intermediate glacial conditions. While our experiments were run with pre-industrial topography, sea level and wind fields, the initial location of convection sites between Greenland and the British Islands (areas with lowest density differences over upper 1000 m in Fig. SI.8) resembles the LGM and intermediate glacial circulation states in Ando and Oka (2021) and Zhang et al. (2017)."

2. Use Fig. 3 to help explain the mechanism. For example, it would be more convincing for me if the authors explain the mechanism in the following manner "reduction of radiative forcing first weakens the convection in the Labrador Sea (Fig. 3) by increasing transport of sea ice from the arctic and by reducing the northward heat transport (Fig. 4). However, intensified surface cooling initiates the deepwater formation close to UK (Fig. 3), causing a shift of the AMOC into the second phase. Further reduction in radiative forcing …." Obviously this is not a perfect example but please consider modifying the manuscript in this way.

*When revising the manuscript we refer more to the figures, as suggested. We also clarify our description of processes (see new text in the answer to the previous comment).*

3. Relation of heat transport and the AMOC is alway tricky. They vary together and also the heat transport can either weaken or strengthen the AMOC depending on the background condition (e.g. Paul and Schulz 2002, https://doi.org/10.1007/978-3-662-04965-5_5, Ando and Oka 2021, GRL). Please cite these paper when discussing the effect of heat transport on AMOC and explain why it should work in that sense.

*We add to the discussion of processes changing AMOC strength in our simulations, and cite Ando and Oka, 2021. The revised text in section 3.2 reads:*

*"The simulated step changes in AMOC strength in our simulations are thus the response to gradual surface cooling and freshening, and occur when NADW formation shifts southwards. The resulting redistributions of heat and salinity cause sudden shifts in the vertical density profiles and sea ice expansion which consolidate the new circulation state (Ando and Oka, 2021). In particular, reduced advection of heat and salinity into former locations of deep water formation result in a more stable local water column (Fig. SI.7-9)."*

Experimental setup: I think the authors need to explain why they decide to vary the magnitude of the dust related radiative forcing but not others in their sensitivity experiments (I'm not saying that's bad!). I don't fully understand how this model works, but isn't there another way to do similar experiments, e.g. changing the magnitude of the emissivity of the atmosphere or the magnitude of the ice sheet related radiative forcing? Effect of dust forcing is of course uncertain, but so are others (Tierney et al. 2020).
Related to 2, another question I have is that "Does the radiative forcing by dust affect the global and local temperatures in the same way as the GHG do in this model?" Looking at results from GCMs (e.g. Kawamura et al. 2017 Science Advances, Ohgaito et al. 2018 CP), it is shown that GHG and dust affect the local temperatures in a different way. This information is important especially when we want to use the insight from this study to better understand results of AOGCMs.

*We clarify our methods and specifically note that our applied forcing of radiation reductions are spatially uniform. As such, the pattern of the additional radiative forcing that we prescribe is slightly different to that of GHG. GCM simulations showed that spatially different forcings lead to a very similar temperature pattern due to feedbacks (Boer, G. and Yu, B., 2003. Climate sensitivity and response. Climate Dynamics, 20, pp.415-429.). In either case, our simulations contain the radiative effect of GHG and the additional, uniform 'dust' forcing. Hence, we do not specifically test the temperature effect of dust load changes specifically, but more generically of changes of the atmospheric radiation balance. There might be other ways of implementing this but in our model the effect would be virtually the same. We add the following sentences to our Methods for clarification:*

*"In our experiments, we applied spatially-uniform radiative forcings, to account for uncertain atmospheric optical depth changes due to changes in aerosols and dust, in addition to the better constrained temperature changes due to orbital changes and greenhouse gases, hence termed dust forcing. The scaling of this forcing varies between the simulations and transiently within each simulation."*

"It is important to note that we only consider the radiative effect of an assumed uniform distribution of aerosols in our simulations. In reality, this distribution would be non-uniform and aerosols would have additional effects on atmospheric freshwater fluxes, two factors which are both relevant for AMOC stability (Menary et al., 2013) but are poorly constrained for the last 780 kyr."

NADW formation in Norwegian/Greenland sea: This might be related to comments from other reviewers, but some previous studies have suggested the importance of cessation/resumption of convection over the Norwegian Sea when considering the thermal threshold of the AMOC (Oka et al. 2012). Please describe this feature in the Introduction and add some discussion wherever appropriate.

*In the pre-industrial model state of Bern3D deep water formation does not occur north of the Irminger Sea. In the revised manuscript, we discuss the importance of the location of deep water formation sites at the beginning of the experiment for the existence of thermal thresholds. Specifically, we will add the following paragraph to section 3.3:*

"The primary importance of salinity and heat redistributions as well as sea ice extent in the North Atlantic for the simulated AMOC shifts resembles the findings from Ando and Oka (2021)'s hosing experiments under LGM conditions and Zhang et al. (2017)'s simulations of AMOC shifts in response to $CO_2$ changes in intermediate glacial conditions. While our experiments are run with pre-industrial boundary conditions, the initial location of convection sites between Greenland and the British Islands (areas with lowest density differences over upper 1000m in Fig. SI.8) resembles the LGM and intermediate glacial circulation states in Ando and Oka (2021) and Zhang et al. (2017). Ganopolski and Rahmstorf (2001) found that the possibility of a southward shift of deep convection depends on the latitude of prior deep convection and the density field further south, and Oka et al. (2012) showed that the location of deep convection and its distance from the winter sea ice edge shape thermal thresholds in AMOC strength."

Specific comments:

P1L25-26: Given the limitations in the model, I think it would be safe to add "in this model" at the end of the sentence.

*We specify that these results are only valid for our model in the revised manuscript.*

P1L31-33: Isn't this the other way round; relatively salty water gets cooled by the atmosphere, the vertical density gradient weakens, and the water sinks and forms the NADW.

*The reviewer is correct. We amend the sentence so that it reads:*

"The Atlantic Meridional Overturning Circulation (AMOC) transports warm waters from the Southern Hemisphere and the Mexican Gulf towards the Nordic Seas, until the gradually cooled salty water sinks after losing enough buoyancy and forms North Atlantic Deep Water (NADW)."

P2L34: Perhaps "sensitive" -> "dependent"?

*We change the wording as suggested.*

P3L13-25: So many references are missing in this paragraph. Please add the appropriate reference for each sentence. (e.g. references for Bern3D model, references for freshwater flux corrections)

*We will add the references describing the details of the Bern3D model and its setup as suggested.*

P4L11-13: How did you define the maximum ice extent? Is it from the LGM?

*We clarify our definition of the maximum forcing. Specifically, we change our description in the Methods section to the following:*

"The maximum radiative dust forcing, defined via the peak LGM value in the smoothed δ18O stack, is a free parameter, ranging from 0 to -8 W/m$^2$ relative to PI (Simulations A.0 to A.8)"

P5L23-24: Better to say "stable"->"monostable", "unstable"->"bistable" here.

*With our stability tests, we assessed how resilient the circulation is to a small perturbation, i.e. whether it is close to a potential bifurcation point. However, we did not test each circulation state for mono- or bistability and the existence of bifurcation points. Hence, we do not think that the suggested terminology is appropriate at this point in the manuscript. Instead, we will improve our terminology for our stability tests to clarify as follows:*

"Our stability experiments demonstrate that the circulation modes before and after the shifts recover from small freshwater perturbations, and can thus be considered as stable, i.e. sufficiently far from bifurcation points to recover from the small perturbation"

P11L11-15: I could not understand this sentence. Can you further elaborate on this, please?

*We rewrite the description of simulation B.slow with a clearer discussion of the relevant processes. We describe changes at the beginning of the simulation as follows:*

"Initially, the whole Atlantic surface ocean cools and freshens, leaving the meridional temperature and salinity gradients almost unchanged (Fig 5e). However, NADW becomes less salty and colder in consequence (not shown) and the vertical density profiles in the subpolar North Atlantic change due to the temperature and salinity changes (Fig. SI.7-8)."

P13L4-5: Not quite sure what this positive feedback means here. In general, a surface cooling will reduce the SST and hence increase the surface density while the cooler SST reduces evaporation and causes a reduction in surface salinity and surface density. So isn't it a negative feedback?

*We apologise for the confusion and the poorly formulated paragraph. We rewrite the paragraph for a clearer discussion of the relevant processes. Instead of feedbacks, we write of stabilising processes, which is a clearer terminology. The new paragraph is:*

"The resulting redistributions of heat and salinity cause sudden shifts in the vertical density profiles and sea ice expansion which consolidate the new circulation state. The downwelling zones are sensitive to heat and salt flux changes, because any reduction in sea surface temperatures (SST) increases surface density but simultaneously reduces evaporation in ice-free areas, thus effectively creating a small freshwater forcing and a negative feedback to the buoyancy changes caused by the initial SST decrease. Sea ice covering the downwelling areas stabilise the water column by

preventing surface ocean cooling and evaporation. The progressive influx of AABW into the North Atlantic is a further process stabilising new circulation states by stratifying the water column from below (Buizert and Schmittner, 2015).. The difference between freshwater transport into the South Atlantic at 32°S and into the Arctic at 62.5°N in Fig. 5f can be used as a measure for the basin-wide salinity feedback (Rahmstorf, 1996, de Vries and Weber, 2005). In our simulation, changes in this metric are predominantly caused by changes in the transport across the northern edge, since transport into the South Atlantic remains almost unchanged throughout the cooling phase of B.slow. North Atlantic salinity is instead governed by changing transport from the subtropics into the North Atlantic and between the North Atlantic and Arctic. As such, the processes involved in the sudden AMOC strength changes, namely density changes in the upper water column, and those that stabilise new circulation states (salinity and heat redistributions, sea ice expansion) mostly operate in the North Atlantic region."

P13L6: Isn't the sea ice feedback a positive feedback?

*This is again an unclear description which we revise. In the new version, we mention sea ice expansion as a stabilising process.*

Figs.2 and 9: Very nice figures.

*Thank you!*

References

[revised manuscript text omitted]

---

## Author Comment (AC3)

*We thank the reviewer for their time and effort, and the constructive comments, which helped to improve our manuscript.*
*Below are our detailed point-by-point replies and suggested manuscript improvements (blue) for each comment (black).*

Minor Comments:

Page 2:

l.3-4: This is the first time future AMOC stability is mentioned. It may be worthwhile to add a few sentences linking past and future AMOC stability.

*We agree that the mention of future AMOC stability at this point is not well-connected to the rest of our manuscript. Since our study is only concerned with AMOC stability at pre-industrial and colder temperatures, we remove the sentences on future AMOC stability.*

l.45: It would also be helpful to provide a bit more context on thermal thresholds. The previous two paragraphs mostly talk about the haline part (i.e. surface freshwater input and salinity redistribution). Which models have been used to analyse thermal AMOC thresholds and for which climate states? And could you comment on whether the AMOC in intermediate complexity models tends to be more or less or similarly stable as in fully coupled earth system models (e.g. the AMOC in ocean-only models is known to be more prone to instabilities than in coupled GCMs).

*We add more information about the models used in studies investigating thermal forcings on AMOC, and provide more references. Further, we now explicitly mention that bistability of AMOC under thermal forcing has been observed in both coupled and uncoupled GCMs. The updated paragraph in the introduction will read:*

*"Such possible circulation state shifts were first identified in box models (Stommel 1961) and confirmed in intermediate complexity models and global circulation models (Jackson and Wood, 2018, review in Jackson et al, 2023). Systematic testing of AMOC stability is done more easily in lower complexity models than General Circulation Models (GCMs), but the existence of multiple AMOC equilibria seems to be determined by the model-dependent existence and strength of feedbacks, with more complex models including more feedbacks that might change AMOC stability (Weijer et al., 2019)"*

*"Besides salinity changes, numerical experiments with GCMs also show that the vertical temperature profile affects AMOC stability (Haskins et al., 2020). Short-term AMOC weakening in response to warming has been simulated by a wide range of GCMs (e.g. Mikolajewicz et al., 1990, Gregory et al., 2005, Weijer et al., 2020). Thermal forcing of the North Atlantic has also been found to cause longer term gradual changes in AMOC strength in intermediate and higher resolution models (Knorr and Lohmann, 2007, Zhang et al., 2017, Galbraith and Lavergne, 2019). In addition, bistability of AMOC under thermal forcing has been found in uncoupled and coupled GCMs (Oka et al., 2012, Klockmann et al., 2018), and thermal forcing, especially of the Southern Ocean, can cause abrupt AMOC state transitions similar to freshwater hosing in the North Atlantic (Oka et al., 2021, Sherriff-Tadano et al., 2023)."*

Page 4:

l.16-17: Can you briefly explain why it is a useful approximation to use the LR04 stack as a scaling for the dust radiative forcing?

*We now mention the close correlation of reconstructed dust fluxes with ice volume and provide a reference in the revised manuscript. The new text reads:*

*"The LR04 stack was chosen because it is the only complete record with constant temporal resolution over the simulated period. In our experiments, we applied spatially-uniform radiative forcings, to account for uncertain atmospheric optical depth changes due to changes in aerosols and dust, in addition to the better constrained temperature changes due to orbital changes and greenhouse gases, hence termed dust forcing. The scale of this forcing varies between the simulations. The maximum radiative dust forcing, defined to occur at the LGM, is a free parameter, ranging from 0 to -8 W/m2 relative to PI (Simulations A.0 to A.8). To construct the forcing, we scaled the maximum forcing linearly with the smoothed LR04 stack, given the close correlation of reconstructed dust fluxes and ice volume likely due to the dominant role of wind field, sea level and hydrological cycle on dust fluxes (Winckler et al., 2008)"*

Page6:

Fig1. Could you also show the combined radiative forcing of all three forcings? That would make it easier to identify periods of changing radiative forcing.

*We add the combination of our dust forcing and the radiative forcing from greenhouse gases to Fig. 1. The orbital forcing does not cause substantial variations of the global radiation balance but rather the spatial and seasonal distribution of insolation. Hence, we prefer to keep showing the insolation changes at 65°N separately as an indication of how high-latitude radiative forcing evolved over the last glacial cycle.*

Page7:

l.24-26: How do you assess stability here?

*AMOC stability is a key concept for our study. However, we have not objectively defined it here, as the varying boundary conditions make it difficult to define an objective criterion that identifies the AMOC stability correctly in both full interglacial and glacial states. Instead, we chose to refer to 'stable modes', which are modes that are occupied by the AMOC most often, as diagnosed from Fig. 2. We agree that this is misleading and remove mentioning of stability from this paragraph, writing instead about the frequency of occurrence.*

Page8:

Fig.3 could you rotate the maps in the upper panels by 45°, so that the perspective on the North Atlantic becomes more easily comparable to the lower panels?

*We change the maps to focus on the North Atlantic region specifically.*

l.22 and 26: Do you show stratification? The lower panles of Fig.3 only show surface density changes. Would it make sense to show stratification? Or do you infer increased stratification simply because of the lighter surface waters?

*We change the figure to include panels that show the density difference across the upper 1000 m of the water column as a metric for stratification.*

Page 9:

l.5-10: I found this paragraph difficult to read and follow. If none of the differences is statistically significant, would it not be sufficient to report that MBT has no statistically significant effect on the AMOC response?

*We shorten this paragraph by removing details about the differences, only mentioning their non-significance, as suggested by the reviewer.*

Page 11:

l.16-19: Does this refer to Fig.5 d? And in general: more specific references to Fig.5 could be made though out this page, to make it easier to follow. It is not always clear  whether the text on this page refers to Fig.5, some other Figure or to results not shown.

*We reference figures more explicitly for the description of the processes.*

l.20-21: Can you name the two processes and timescales explicitly? I guess they are N.Atl. freshwater changes (fast) and AABW propagation (slow), but it would be good to have them spelled out.

*Yes, these were the processes we referred to. However, changes in the North Atlantic are more relevant for the observed AMOC changes, AABW propagation seems to have more of a stabilising rather than destabilising effect. We remove this sentence and instead discuss changes in the North Atlantic and Southern Ocean separately.*

Page 12:

l.20-24: I do not really see the further reduction in NADW export. To me, the distributions of all three water masses look almost identical at 23 and 24.5 kyr. Also, I do not really see NADW replacing AAIW, the upper NADW boundary does not seem to change and if anything, the southward extent of NADW also decreases.

*We agree, our descriptions here were not accurate. We clarify this as follows:*

*"The first abrupt shift in AMOC strength at 24.5 kyr in B.slow has only small effects on the water mass distribution. It mainly leads to a reduced concentration of NADW at intermediate depths of the North Atlantic >45°N and a small increase of AABW concentration in the abyssal North Atlantic (Fig. 5d)."*

Page 13:

l.25: What about the strong variance at 6kyr?

*The strong variance at 6 kyr is associated with density changes in the North Atlantic. However, the AMOC appears to not undergo a state transition during this time. We add a description of this to the discussion of simulation B.slow to section 3.2 as follows:*

"Initially, the whole Atlantic surface ocean cools and freshens, leaving the meridional temperature and salinity gradients almost unchanged (Fig 5e). However, NADW becomes less salty and colder as a consequence of the changes in the surface ocean (not shown) and the vertical density profiles in the subpolar North Atlantic steepen due to the temperature and salinity changes (Fig. SI.7-8). After about 6 kyr, the changes in the North Atlantic density profile result in shifts in the spatial pattern of NADW formation. NADW formation moves south as vertical density profiles in the subpolar east North Atlantic stabilise under a freshening of the surface and density profiles further south steepen due to surface cooling combined with subsurface warming (Fig. SI.7-9). These changes do not cause a step-change in AMOC strength,but freshwater and heat advection into the North Atlantic is reduced, sea ice expansion increases in the eastern North Atlantic and AMOC variance (calculated over a moving 50-year window) is increased (Fig. 5). Transport of heat and salinity into the North Atlantic decreases (Fig. 5f, g) and North Atlantic SST and SSS decrease (Fig. 5e). Reduced influx of subtropical surface waters also causes sudden cooling and freshening in the Irminger Sea (Fig. SI.8)."

Page 14:

l.32-40: Is this part meant in contrast to other models? The last sentence is also almost impossible to follow. Please consider a clearer formulation.

*We rewrite this paragraph for a clearer discussion. The new text in section 3.3 is:*

"In a model with a dynamic energy moisture balance component, atmospheric cooling reduces evaporation and the water-holding capacity. With this feedback enabled, cooling can then affect seawater density directly via changing temperatures, and indirectly via changing the meteoric freshwater balance and surface salinities. These changes would induce additional kinematic changes (i.e., in the wind fields) in fully dynamic atmosphere models, but are kept constant in our simulations, i.e. the moisture content of air changes with climate but not the direction or strength of winds which disperse it. In our model, a decrease in the water-holding capacity of air therefore directly leads to a reduction of the large scale atmospheric moisture transport from low to high latitudes. Accordingly, wind stress fields are also kept constant here. Changes in wind stress have been documented to exert important controls on AMOC stability (e.g. Arzel et al., 2008, Yang et al., 2016) and thermal thresholds (Oka et al., 2012). These effects have been investigated in detail with the Bern3D model by Pöppelmeier et al. (2021) focusing on LGM boundary conditions.

Page 16/17:

Meta stable AMOC modes: How are the excitable/metastable states defined? By increased AMOC variance as in Fig.5? How do the metastable states relate to the four AMOC states I-IV from the beginning? Also, please consider adding the corresponding kyrs behind MIS3/4/5e etc, so that it is

easier to identify the right parts of Fig.9 for those readers who do not have those numbers at the top of their heads.

*We define excitable states as times when AMOC adopts intermediate modes II and III, which show more frequent AMOC strength shifts than the interglacial and glacial modes I and IV, respectively. We also add the requested age information. The new text in section 3.4 is:*

"Finally, we can test whether our simulations capture the periods with increased frequency of AMOC transitions that are indicated by proxies over the last eight glacial cycles. Using our 788 kyr long simulations in simulation set A, we determined when the radiative forcing pushed the AMOC into 'excitable' circulation modes, i.e. modes II and III, which show more frequent AMOC strength shifts than the interglacial and glacial modes I and IV (Fig. 1 and SI.2), and how this varied with the applied forcing strength (Fig. 9)."

Page 18:

l.24-27: This would be very interesting indeed. I look forward to the follow-up :)

*We too!*

Technical/Editorial Comments:

l.34: delete "boundary" after "Atlantic"

Deleted.

l.36-42: Very long and hard to read sentence. Consider reformulating for better readability. Also: which climate is being referred to at the end of the sentence? Probably North Pacific climate but it is not immediately clear.

*We simplify and clarify this sentence as follows:*

"It also affects global climate by shifting the Intertropical Convergence Zone (ITCZ) and monsoon systems (Wang et al., 2001, Bozbiyik et al, 2011), and interacting with the regional climate and deep water formation in the North Pacific (Okazaki et al., 2010, Menviel et al., 2012, Praetorius and Mix, 2014)."

l.43-47: same as comment above. Also: Does the last half sentence ("and by modulating atmospheric greenhouse gas concentrations") still correctly belong to the beginning of the sentence ("It influences deep ocean nutrient and oxygen concentrations")?

*We rewrite this sentence and clarify the role for greenhouse gas concentrations. The new sentences read:*

"The AMOC furthermore shapes biological surface productivity by regulating nutrient supply to the surface ocean in the Atlantic and Pacific (Tetard et al., 2017, Joos et al., 2017). On its southward

path in the Atlantic, it influences deep ocean nutrient, carbon, and oxygen concentrations (Broecker, 1991). By affecting primary production and deep ocean carbon storage, AMOC changes also modulate atmospheric greenhouse gas concentrations (e.g. Menviel et al., 2008)."

Page 2:

l.2: "which had regional [...]" instead of "and had regional [...]"

We amend this according to the reviewer's suggestion.

Page 4:

l.13: What is meant with "rest of the past 800kyr"? Rest with respect to what? The spin up state?

*This is a leftover from an amended sentence of a previous manuscript version. We remove "rest of the".*

Page 10:

Fig.5: Please increase the font size for better readability.

*We increase the font size as suggested.*

Page 12:

Fig.6: Please increase the font size for better readability.
*We increase the font size as suggested.*

Page 14:

l.23: The name of the ocean model is COCO (the ocean component of MIROC)

*We amend the sentence accordingly.*

Page 15:

Fig.7: Please increase the font size for better readability.

*We increase the font size as suggested.*

Page 16:

l.28: wrong Figure reference? Should be Figure 1?

*That is correct, we change the figure reference accordingly to Fig. 1 in the revised manuscript.*

Page 18:

l.14: delete "but"

Deleted.

References

[revised manuscript text omitted]

---

## Author Response (AR1)

*Dear Editor,*

*We thank you for sending our manuscript for review and giving us this opportunity to reply to the review comments. The comments were constructive and mostly asked for additional citations, clarifications and rephrasing of some text passages and improvements to the figures. We will provide an improved manuscript, addressing all review comments. Below are our detailed point-by-point replies and suggested manuscript improvements (blue) for each comment (black).*

*In the name of all co-authors,*
*Markus Adloff*
* * *
Reviewer 1:

Markus Adloff and colleagues assess the sensitivity of the AMOC to changes in radiative forcing in the intermediate complexity model Bern3D. The range of the radiative forcing is representative of the last 800 kyr. The radiative forcing comprises orbital forcing, greenhouse gases, ice-sheet induced albedo changes and dust forcing. The strength of the radiative forcing is scaled by the maximum dust forcing at the LGM. The authors identify four stable AMOC states, a strong interglacial state, a weak glacial state and two less stable intermediate states. The magnitude of the radiative forcing determines the time the AMOC spends in each of the respective states. The authors analyse the characteristics of the AMOC states and assess the underlying mechanisms through further more idealised simulations. Comparison with available proxy data for sea surface temperature, AMOC strength and climate variability indicate that the simulations contain realistic AMOC behaviour (depending on the forcing strength) and that valuable insights on thermal AMOC thresholds throughout the glacial cycles can be obtained from them.

Overall the paper is of high quality, well written and definitely of interest for a wide audience in the CP community and beyond. Testing for thermal AMOC thresholds in itself is not new, but the length of the simulations and the large covered range of forcing scenarios that can only be achieved through the intermediate complexity of Bern3D provide enough novel insights.

My comments are mostly minor, asking for more clarification or context. I recommend publication after minor revisions.

*We thank the reviewer for their time and effort, and the constructive comments, which helped to improve our manuscript.*
*Below are our detailed point-by-point replies and manuscript improvements (blue) for each comment (black). Line numbers refer to the new manuscript version without track changes.*

Minor Comments:

Page 2:

l.3-4: This is the first time future AMOC stability is mentioned. It may be worthwhile to add a few sentences linking past and future AMOC stability.

*We agree that the mention of future AMOC stability at this point is not well-connected to the rest of our manuscript. Since our study is only concerned with AMOC stability at pre-industrial and colder temperatures, we removed the sentences on future AMOC stability.*

l.45: It would also be helpful to provide a bit more context on thermal thresholds. The previous two paragraphs mostly talk about the haline part (i.e. surface freshwater input and salinity redistribution). Which models have been used to analyse thermal AMOC thresholds and for which climate states? And could you comment on whether the AMOC in intermediate complexity models tends to be more or less or similarly stable as in fully coupled earth system models (e.g. the AMOC in ocean-only models is known to be more prone to instabilities than in coupled GCMs).

*We added more information about the models used in studies investigating thermal forcings on AMOC, and provided more references (lines 95-109, 564-571). Further, we now explicitly mention that bistability of AMOC under thermal forcing has been observed in both coupled and uncoupled GCMs. The updated paragraph in the introduction will read:*

Page 4:

l.16-17: Can you briefly explain why it is a useful approximation to use the LR04 stack as a scaling for the dust radiative forcing?

*We now mention the close correlation of reconstructed dust fluxes with ice volume and provide a reference in the revised manuscript (lines 167-170).*

Page6:

Fig1. Could you also show the combined radiative forcing of all three forcings? That would make it easier to identify periods of changing radiative forcing.

*We added the combination of our dust forcing and the radiative forcing from greenhouse gases to Fig. 1. The orbital forcing does not cause substantial variations of the global radiation balance but rather the spatial and seasonal distribution of insolation. Hence, we preferred to keep showing the insolation changes at 65°N separately as an indication of how high-latitude radiative forcing evolved over the last glacial cycle.*

Page7:

l.24-26: How do you assess stability here?

*AMOC stability is a key concept for our study. However, we have not objectively defined it here, as the varying boundary conditions make it difficult to define an objective criterion that identifies the AMOC stability correctly in both full interglacial and glacial states. Instead, in the old version we chose to refer to 'stable modes', which are modes that are occupied by the AMOC most often, as diagnosed from Fig. 2. We agree that this is misleading and remove mentioning of stability from this paragraph, writing instead about the frequency of occurrence.*

Page8:

Fig.3 could you rotate the maps in the upper panels by 45°, so that the perspective on the North Atlantic becomes more easily comparable to the lower panels?

*We changed the maps to focus on the North Atlantic region specifically.*

l.22 and 26: Do you show stratification? The lower panles of Fig.3 only show surface density changes. Would it make sense to show stratification? Or do you infer increased stratification simply because of the lighter surface waters?

*We changed the figure to include panels that show the density difference across the upper 1000 m of the water column as a metric for stratification.*

Page 9:

l.5-10: I found this paragraph difficult to read and follow. If none of the differences is statistically significant, would it not be sufficient to report that MBT has no statistically significant effect on the AMOC response?

*We shortened this paragraph by removing details about the differences, only mentioning their non-significance, as suggested by the reviewer (lines 316-321).*

Page 11:

l.16-19: Does this refer to Fig.5 d? And in general: more specific references to Fig.5 could be made though out this page, to make it easier to follow. It is not always clear  whether the text on this page refers to Fig.5, some other Figure or to results not shown.

*We referenced figures more explicitly for the description of the processes.*

l.20-21: Can you name the two processes and timescales explicitly? I guess they are N.Atl. freshwater changes (fast) and AABW propagation (slow), but it would be good to have them spelled out.

*Yes, these were the processes we referred to. However, changes in the North Atlantic are more relevant for the observed AMOC changes, AABW propagation seems to have more of*

*a stabilising rather than destabilising effect. We removed this sentence and instead discuss changes in the North Atlantic and Southern Ocean separately.*

Page 12:

l.20-24: I do not really see the further reduction in NADW export. To me, the distributions of all three water masses look almost identical at 23 and 24.5 kyr. Also, I do not really see NADW replacing AAIW, the upper NADW boundary does not seem to change and if anything, the southward extent of NADW also decreases.

*We agree, our descriptions here were not accurate. We clarified this as follows (lines 439-442):*

*"The first abrupt shift in AMOC strength at 24.5 kyr in B.slow had only small effects on the water mass distribution. It mainly led to a reduced concentration of NADW at intermediate depths of the North Atlantic >45°N and a small increase of AABW concentration in the abyssal North Atlantic (Fig. 5d)."*

Page 13:

l.25: What about the strong variance at 6kyr?

*The strong variance at 6 kyr is associated with density changes in the North Atlantic. However, the AMOC appears to not undergo a state transition during this time. We added a description of this to the discussion of simulation B.slow to section 3.2 (lines 383-401).*

Page 14:

l.32-40: Is this part meant in contrast to other models? The last sentence is also almost impossible to follow. Please consider a clearer formulation.

*We rewrote this paragraph for a clearer discussion (lines 549-562).*

Page 16/17:

Meta stable AMOC modes: How are the excitable/metastable states defined? By increased AMOC variance as in Fig.5? How do the metastable states relate to the four AMOC states I-IV from the beginning? Also, please consider adding the corresponding kyrs behind MIS3/4/5e etc, so that it is easier to identify the right parts of Fig.9 for those readers who do not have those numbers at the top of their heads.

*We define excitable states as times when AMOC adopts intermediate modes II and III, which show more frequent AMOC strength shifts than the interglacial and glacial modes I and IV, respectively. We made this clearer in the text (lines 672-677). We also added the requested age information.*

Page 18:

l.24-27: This would be very interesting indeed. I look forward to the follow-up :)

*We too!*

Technical/Editorial Comments:

l.34: delete "boundary" after "Atlantic"

Deleted.

l.36-42: Very long and hard to read sentence. Consider reformulating for better readability. Also: which climate is being referred to at the end of the sentence? Probably North Pacific climate but it is not immediately clear.

*We simplified and clarified this sentence as follows (lines 39-43):*

"It also affects global climate by shifting the Intertropical Convergence Zone (ITCZ) and monsoon systems (Wang et al., 2001, Bozbiyik et al, 2011), and interacting with the regional climate and deep water formation in the North Pacific (Okazaki et al., 2010, Menviel et al., 2012, Praetorius and Mix, 2014)."

l.43-47: same as comment above. Also: Does the last half sentence ("and by modulating atmospheric greenhouse gas concentrations") still correctly belong to the beginning of the sentence ("It influences deep ocean nutrient and oxygen concentrations")?

*We rewrote this sentence and clarified the role for greenhouse gas concentrations (lines 43-48). The new sentences read:*

"The AMOC furthermore shapes biological surface productivity by regulating nutrient supply to the surface ocean in the Atlantic and Pacific (Tetard et al., 2017, Joos et al., 2017). On its southward path in the Atlantic, it influences deep ocean nutrient, carbon, and oxygen concentrations (Broecker, 1991). By affecting primary production and deep ocean carbon storage, AMOC changes also modulate atmospheric greenhouse gas concentrations (e.g. Menviel et al., 2008)."

Page 2:

l.2: "which had regional [...]" instead of "and had regional [...]"

Done.

Page 4:

l.13: What is meant with "rest of the past 800kyr"? Rest with respect to what? The spin up state?

*This is a leftover from an amended sentence of a previous manuscript version. We removed "rest of the".*

Page 10:

Fig.5: Please increase the font size for better readability.

*Done.*

Page 12:

Fig.6: Please increase the font size for better readability.

*Done.*

Page 14:

l.23: The name of the ocean model is COCO (the ocean component of MIROC)

*Done.*

Page 15:

Fig.7: Please increase the font size for better readability.

*Done.*

Page 16:

l.28: wrong Figure reference? Should be Figure 1?

*That is correct, we changed the figure reference accordingly to Fig. 1 in the revised manuscript.*

Page 18:

l.14: delete "but"

Deleted.
* * *
Reviewer 2:

Summary

This study investigates the thermal thresholds of AMOC over the last 800kyr in the Bern3D climate model. By controlling the amplitude of the global cooling via dust forcing, they extract the timing of occurrences of the thermal threshold of AMOC. In addition, they manage to extract four stable and meta-stable AMOC modes, which differ in the locations of NADW formation regions and sea ice extent over the North Atlantic. The paper further explores mechanisms causing the AMOC weakening due to the reduction in the radiative forcing by means of hysteresis experiments. At the end of the paper, the authors provide a discussion on the limitations of the study by not including the orographic effect of the ice sheet in their simulations.

The experiments and results presented in this paper are very interesting (especially figs 2 and 9!) and are of interest of readers of the Climate of the Past. Therefore, I think this paper should be published. However, the explanation of the mechanism of the AMOC changes appears unclear to me (also noted by other reviewers). Additionally, there are some ambiguity in the experimental setup. Addressing these points would improve the overall quality of the manuscript. The comments are summarized below.

*We thank the reviewer for the invested time in evaluating our study and the thoughtful comments that have helped to substantially improve the manuscript.*
*Below are our detailed point-by-point replies and suggested manuscript improvements (blue) for each comment (black). Line numbers refer to the new manuscript version without track changes.*

Main comments:

Mechanisms by which the reduction of radiative forcing weakens the AMOC: Other reviewers have pointed out many points, so I'll just list potential ways to improve the manuscript.

1. Separate the paragraph explaining the effect of the Southern Ocean and North Atlantic. Section 3.2 goes back and forth between the role of NA and Southern Ocean. This makes it hard to follow the discussion. Related to this, Buizert and Schmittner (2015) provides a nice summary on the role of Southern Ocean. Ando and Oka (2021, GRL) also gives useful insight on the role of sea ice and heat transport on the stability of the AMOC. These two studies further performed hysteresis experiments with freshwater forcing. While the way of hysteresis experiment is not the same as in this study, I feel these studies should be cited and included in the discussion of the mechanism.

*We rewrote the description of the processes at play in simulation B.slow (lines 370-419, 450-480), and followed the reviewer's advice to discuss changes in the North Atlantic and*

*Southern Ocean separately. We now also refer to Buizert and Schmittner (2015) and Ando and Oka (2021) in the discussion of our results (lines 564-599).*

2. Use Fig. 3 to help explain the mechanism. For example, it would be more convincing for me if the authors explain the mechanism in the following manner "reduction of radiative forcing first weakens the convection in the Labrador Sea (Fig. 3) by increasing transport of sea ice from the arctic and by reducing the northward heat transport (Fig. 4). However, intensified surface cooling initiates the deepwater formation close to UK (Fig. 3), causing a shift of the AMOC into the second phase. Further reduction in radiative forcing ...." Obviously this is not a perfect example but please consider modifying the manuscript in this way.

*We added more references to the figures, as suggested. We also clarified our description of processes (lines 366-504).*

3. Relation of heat transport and the AMOC is alway tricky. They vary together and also the heat transport can either weaken or strengthen the AMOC depending on the background condition (e.g. Paul and Schulz 2002, https://doi.org/10.1007/978-3-662-04965-5_5, Ando and Oka 2021, GRL). Please cite these paper when discussing the effect of heat transport on AMOC and explain why it should work in that sense.

*We added to the discussion of processes changing AMOC strength in our simulations, and cited Ando and Oka, 2021 (lines 457-463):*

*"The simulated step changes in AMOC strength in our simulations were thus the response to gradual surface cooling and freshening, and occurred when NADW formation shifted southwards. The resulting redistributions of heat and salinity caused sudden shifts in the vertical density profiles and sea ice expansion which consolidated the new circulation mode (Ando and Oka, 2021). In particular, reduced advection of heat and salinity into former locations of deep water formation resulted in a more stable local water column (Fig. SI.7-9)."*

Experimental setup: I think the authors need to explain why they decide to vary the magnitude of the dust related radiative forcing but not others in their sensitivity experiments (I'm not saying that's bad!). I don't fully understand how this model works, but isn't there another way to do similar experiments, e.g. changing the magnitude of the emissivity of the atmosphere or the magnitude of the ice sheet related radiative forcing? Effect of dust forcing is of course uncertain, but so are others (Tierney et al. 2020).
Related to 2, another question I have is that "Does the radiative forcing by dust affect the global and local temperatures in the same way as the GHG do in this model?" Looking at results from GCMs (e.g. Kawamura et al. 2017 Science Advances, Ohgaito et al. 2018 CP), it is shown that GHG and dust affect the local temperatures in a different way. This information is important especially when we want to use the insight from this study to better understand results of AOGCMs.

*We clarified our methods and specifically noted that our applied forcing of radiation reductions are spatially uniform. As such, the pattern of the additional radiative forcing that*

*we prescribed is slightly different to that of GHG. GCM simulations showed that spatially different forcings lead to a very similar temperature pattern due to feedbacks (Boer, G. and Yu, B., 2003. Climate sensitivity and response. Climate Dynamics, 20, pp.415-429.). In either case, our simulations contain the radiative effect of GHG and the additional, uniform 'dust' forcing. Hence, we do not test the temperature effect of dust load changes specifically, but more generically of changes of the atmospheric radiation balance. There might be other ways of implementing this but in our model the effect would be virtually the same. We clarified our method description accordingly (lines 160-164, 182-186).*

NADW formation in Norwegian/Greenland sea: This might be related to comments from other reviewers, but some previous studies have suggested the importance of cessation/resumption of convection over the Norwegian Sea when considering the thermal threshold of the AMOC (Oka et al. 2012). Please describe this feature in the Introduction and add some discussion wherever appropriate.

*In the pre-industrial model state of Bern3D deep water formation does not occur north of the Irminger Sea. In the revised manuscript, we discuss the importance of the location of deep water formation sites at the beginning of the experiment for the existence of thermal thresholds. We added a discussion of this (lines 104-109, 564-571).*

Specific comments:

P1L25-26: Given the limitations in the model, I think it would be safe to add "in this model" at the end of the sentence.

*We specified that these results are only valid for our model in the revised manuscript.*

P1L31-33: Isn't this the other way round; relatively salty water gets cooled by the atmosphere, the vertical density gradient weakens, and the water sinks and forms the NADW.

*The reviewer is correct. We amended the sentence (lines 30-33):*

*"The Atlantic Meridional Overturning Circulation (AMOC) transports warm waters from the Southern Hemisphere and the Mexican Gulf towards the Nordic Seas, until the gradually cooled salty water lost enough buoyancy and sinks, forming North Atlantic Deep Water (NADW)."*

P2L34: Perhaps "sensitive" -> "dependent"?

*Done.*

P3L13-25: So many references are missing in this paragraph. Please add the appropriate reference for each sentence. (e.g. references for Bern3D model, references for freshwater flux corrections)

*We added the references describing the details of the Bern3D model and its setup as suggested.*

P4L11-13: How did you define the maximum ice extent? Is it from the LGM?

*We clarified our definition of the maximum forcing. Specifically, we changed our description in the Methods section (lines 165-167):*

"The maximum radiative dust forcing, defined via the peak LGM value in the smoothed δ18O stack, is a free parameter, ranging from 0 to -8 W/m$^2$ relative to PI (Simulations A.0 to A.8)"

P5L23-24: Better to say "stable"->"monostable", "unstable"->"bistable" here.

*With our stability tests, we assessed how resilient the circulation is to a small perturbation, i.e. whether it is close to a potential bifurcation point. However, we did not test each circulation state for mono- or bistability and the existence of bifurcation points. Hence, we do not think that the suggested terminology is appropriate at this point in the manuscript. Instead, we improved our terminology for our stability tests to clarify as follows (lines 482-485):*

"Our stability experiments demonstrate that the circulation modes before and after the shifts recover from small freshwater perturbations, and can thus be considered as stable, i.e. sufficiently far from bifurcation points to recover from the small perturbation"

P11L11-15: I could not understand this sentence. Can you further elaborate on this, please?

*We rewrote the description of simulation B.slow with a clearer discussion of the relevant processes. We described changes at the beginning of the simulation (lines 383-388):*

"Initially, the whole Atlantic surface ocean cools and freshens, leaving the meridional temperature and salinity gradients almost unchanged (Fig 5e). However, NADW becomes less salty and colder in consequence (not shown) and the vertical density profiles in the subpolar North Atlantic change due to the temperature and salinity changes (Fig. SI.7-8)."

P13L4-5: Not quite sure what this positive feedback means here. In general, a surface cooling will reduce the SST and hence increase the surface density while the cooler SST reduces evaporation and causes a reduction in surface salinity and surface density. So isn't it a negative feedback?

*We apologise for the confusion and the poorly formulated paragraph. We rewrote the paragraph for a clearer discussion of the relevant processes (lines 459-480). Instead of feedbacks, we wrote of stabilising processes, which is a clearer terminology.*

P13L6: Isn't the sea ice feedback a positive feedback?

*This is again an unclear description which we revised. In the new version, we mention sea ice expansion as a stabilising process (lines 528-536).*

Figs.2 and 9: Very nice figures.

*Thank you!*
* * *
Reviewer 3:

In this manuscript, the authors performing several sets of transient experiments in Bern3D investigate thermally induced AMOC stability across glacial cycles. The results are new and complementary to our current theoretical understanding of glacial abrupt climate change. I believe this is a nice contribution to the community and suitable to Clim Past, but I reserve my recommendation for publication of this version since there remains room to improve its robustness and significance. In general, the authors shall 1) provide a more comprehensive introduction/discussion by considering at least most relevant literatures regarding AMOC stability during glacial cycles, 2) improve the clarity for mechanisms and feedback involved before, during and aOer AMOC transitions and 3) substantiate conclusions/statements by specifying the corresponding plots or adding direct modeling results/literatures. In addition, I would also recommend adding the 800-kyr results at least in the supplementary to provide an overview of the results, which would be of great interest for colleagues who are working on earlier glacial cycles as well.

*We thank the reviewer for their time and effort reviewing our manuscript, and for the constructive comments which have helped to substantially improve our text. We addressed the reviewer's suggested improvements by completely rewriting our description and discussion of the processes that occur in the model, adding results to the figures in the main text and adding new figures to the SI, and referencing these figures more thoroughly in the text. Specifically, we made the changes outlined in our replies to the detailed comments below.*
*Below are our detailed point-by-point replies and suggested manuscript improvements (blue) for each comment (black). Line numbers refer to the new manuscript version without track changes.*

Detailed comments are as follows:

P2L15-18: Freshwater input might be positive feedback to AMOC weakening as well. please refer to Barker et al 2015 and rephrase the sentences accordingly here as well as in L23-24.

*We added the suggested reference and now mention the possibility for freshwater feedbacks. Specifically, we added the following sentence to our introduction (lines 72-74):*

*"Lags between the appearance of ice-rafted debris and the reconstructed cooling, however, suggest that freshwater fluxes could have instead acted as a positive feedback to AMOC weakening rather than triggering it (Barker et al., 2015)."*

P2L25-29: other key relevant paper should be cited, for instance, Zhang et al., 2014, 2017.

*We added references to Zhang et al., 2014 and 2017 and Vettoretti, 2022.*

P2L33: also consider citing Zhang et al 2021, Vettoretti et al 2022 here.

*Done.*

P2L45-47: Please add relevant papers after the first sentence (e.g. Knorr and Lohmann 2007, Zhang et al 2017, Galbraith and de Lavergne 2018, etc.)

*Done.*

P4 L5: one predominant feature of glacial cycle is the development and demise of northern hemisphere ice sheet, involving both area and height, of which impacts on climate system are not the same. The former, as discussed in this study, via its albedo feedback is a thermal impact, while the latter, via its impacts on winds, is a kinetic impact (Zhang et al., 2014). In addition, there is no change in Bering Strait considered as well (Hu et al., 2011) (P5L4, a typo there). I was wondering how far these additional setups can alter the key messages of the thermal thresholds in this study. As seeing in my following comments, at least a comprehensive discussion around this is required.

*We added more discussion of other factors for AMOC stability (lines 584-599), see also our answers to further comments on this topic below.*

P6 3.1: it would be good to present the 800kyr long transient simulation results. In Figure 1, it is of great help to add the radiative forcing curves to enable a comparison with B.slow experiment.

*We added the according figure to the SI (Fig. SI.2).*

P7L15-17: As alluded, lacking feedback from topo changes might overestimate the LGM cooling caused by radiative forcing decrease because higher NHIS can cause a stronger AMOC which promotes heat release from the ocean and hence North Atlantic warming. This might stimulate some discussion perhaps in data-model comparison or model limitation sections.

*We added this to our discussion as suggested (lines 584-599).*

P8. Fig3: given the North Atlantic and Nordic Sea are the key regions for AMOC state shift, it would be better to provide a zoom-in plot for this region, especially for the sea ice fraction

plot. Please also revise the color scheme for "sea ice cover fraction" to highlight change in the low values (<0.5) or just provide anomalous field as delta Density. Please also include lat-lon info in the plots. In addition, as you are discussing AMOC states, AMOC plots are highly recommended in this figure.

*We changed the figure to show sea ice anomalies in the North Atlantic and added coordinates. We also added vector plots showing AMOC circulation to the SI. Further, we added AMOC plots to Fig 2.*

P8L18-20: in the state (II), deep water formation is enhanced in west and south of Greenland. In general, it is more reddish in State (II) than in State (I), but why the AMOC is weakened in the former. Is this due to that convection in the western North Atlantic is not the key to the strength of the AMOC?

*Yes, the mixed layer depth that we diagnosed from the annually-averaged model output is not a good metric to understand changes in AMOC strength. We explain the density changes more explicitly in the new manuscript version, and note that some changes in the locations of downwelling occur without changing AMOC strength (lines 379-395). We removed the plots of mixed layer depth and instead now show the absolute vertical density gradients in each state.*

P8L25: "south-flowing fresh Arctic waters further stratify …". This is a key process to stabilize the glacial AMOC state, but in this version, there is not direct evidence to support it. Note that freshwater convergence in Fig5e cannot provide such support to this statement because it is a sum of freshwater flux across both 40N and 70N in the North Atlantic.

*We now show the freshwater fluxes across each latitude separately in the updated Fig. 5. We also clarified that the spread of cold, fresh surface water in the North Atlantic stabilises the circulation state. We cannot actually determine whether the water comes from the Arctic or just turns more Artic-like due to reduced influx of southern surface water. We rephrased the text accordingly.*

P9L10: what is Kolmogrov-Smirnov test? Add details and reference.

*We provided some more information and a reference (lines 319-321):*

*"none are statistically significant in the two-sided Smirnov test, which determines the likelihood that two distributions are the same (Berger and Zhou, 2014), even at the 50% confidence level"*

P10 Fig 5: Panel e, it would be good to interpret meanings of positive/negative values of freshwater convergence to help readers understand this plot (e.g. positive values indicate freshwater import and hence a stable AMOC). In addition, the definition of freshwater convergence should be added to the Method section. It is worth noting that this AMOC stability indicator (Liu et al., 2014 Clim Dyn) predict a mono-stable AMOC regime in B.slow., in contrast to the hysteresis feature shown in Fig 4b. In addition, comparing the panel a) with

Fig 4b, it appears that B.slow.b is initialized from a AMOC state that is bistable with respect to radiative forcing. If so, why the AMOC recovers to its initial strong mode after removing the freshwater input? Typo in y-axis labels of panel c). it is also good to add radiative forcing panel on the top of it, with a vertical shaded bar to highlight periods when AMOC is bistable.

*We amended Figure 5 as suggested. The hysteresis shown in Fig 4b is not the result of a traditional perturbation experiment with freshwater hosing but is the transient response to the applied radiative forcing. We are not sure if stability with regard to a freshwater perturbation is the same as stability in the face of changing boundary conditions, as caused by the radiative forcing. We are therefore careful with the interpretation of freshwater transport as stability indicators in our study. We added a discussion of freshwater transport to section 3.1 (lines 467-476):*

"The difference between freshwater transport into the South Atlantic at 32°S and into the Arctic at 62.5°N in Fig. 5f can be used as a measure for the basin-wide salinity feedback (Rahmstorf, 1996, de Vries and Weber, 2005). In our simulation, changes in this metric were predominantly caused by changes in the transport across the northern edge, since transport into the South Atlantic remained almost unchanged throughout the cooling phase of B.slow. North Atlantic salinity is instead governed by changing transport from the subtropics into the North Atlantic and between the North Atlantic and Arctic. As such, the processes involved in the sudden AMOC strength changes, namely density changes in the upper water column, and those that stabilised new circulation modes (salinity and heat redistributions, sea ice expansion) mostly operated in the North Atlantic region."

P11L31: how do you identify the reduced heat convergence "off the British Isles" based on the time series in Fig5?

*The geographic information was not derived from the time series but provided as extra information to contextualise the time series. We added a figure to SI showing the discussed spatial pattern (new Fig. SI.11).*

P11L33: It is also not logically clear why this is the cause to the northward spread of AABW. In Fig5, the northward intrusion of AABW is starting from the beginning of the experiment, not lagging the reduction of heat convergence in North Atlantic.

*We agree, and changed the sentence to point out coincidence rather than causality. The experiment is initialised with no AABW tracer in the North Atlantic. Initially, changes in the concentration of the AABW tracer in the North Atlantic are small. Its amount only begins to rise substantially at ~15 kyr and shows the biggest jump at ~27 kyr when the heat convergence also declines.*

P11L35: why "heat advection to >55N stops entirely"? could the authors present the evidence?

*We apologise, this should have been 'heat convergence'. We corrected this and added a figure of the spatial pattern of heat convergence changes to the SI (new Fig. SI.11).*

P11L37-39: Again, no direct lines of evidence to support this statement. Does the contemporary sea ice expansion and its seasonality contribute to the freshening in the eastern Nordic Sea? As well as in P11L42-43. Please clarify. Is there a bipolar thermal seesaw during abrupt AMOC reduction in B.slow? The results appear to show that bipolar sea ice change out of phase with AMOC/NADW change – sea ice expansion with NADW weakening. The subdued thermal seesaw in B.slow indicates the dominant role of decreasing radiative forcing in controlling bipolar change.

*Yes, we thank the reviewer for pointing this out. There is a small bipolar seesaw effect. We mention this explicitly in the revised manuscript. We also added a plot of with spatial patterns of changes in B.slow to the SI and add the following text to section 3.1 (lines 410-415):*

"The biggest AMOC weakening at ~27 kyr is also accompanied by a weak bipolar seesaw effect, which causes a temporary decline in sea ice coverage in the Atlantic sector of the Southern Ocean (Fig. 5). It is, however, too small to reduce the radiation-driven sea ice increase in the longer term."

P12L27: what's the statement "… increased heat advection into the North Atlantic" based on?

*This statement was erroneous and we deleted it. AMOC strength is not constant at the beginning of the experiment but weakens slowly, while the spatial pattern of deep convection and heat convergence in the North Atlantic change. We rewrote our description of the processes responsible for AMOC changes and use more references to figures. We added the spatial patterns of heat convergence changes to the new Fig. SI.11.*

P12L29: weakened north ward transport of what? Upper cell of the AMOC?

*We clarified that we mean the transport of salt and heat.*

P13L1: please show the weakened the meridional salinity gradient in the North Atlantic.

*We apologise, this was meant to say weakened meridional salinity transport, i.e. an increased salinity gradient. We corrected the statement in the text and added the temporal evolution of salinity in the Irminger and Caribbean Seas to Fig. 5 to show the increased meridional salinity gradient.*

P13L3-5: how does the increased surface density promote SST decrease? This is not clear at all here.

*We agree with the reviewer that this formulation was misleading. We meant to express that SST changes have a direct effect on water density and an indirect one via influencing evaporation, and that temperature-driven evaporation changes counteract the buoyancy forcing caused by the temperature change. We rewrote this paragraph as follows (lines 459-466):*

"The deep water formation regions are sensitive to heat and salt flux changes, because any reduction in sea surface temperatures (SST) increases surface density but simultaneously reduces evaporation in ice-free areas, thus effectively creating a small freshwater forcing and a negative feedback to the buoyancy changes caused by the initial SST decrease. Sea ice covering the downwelling areas stabilises the water column by preventing surface ocean cooling and evaporation. The progressive influx of AABW into the North Atlantic is a further process stabilising new circulation modes by stratifying the water column from below (Buizert and Schmittner, 2015)."

P13L5-7: the authors proposed that sea ice expansion over convection sites acts as negative feedback in response to SST cooling, which is not convincing. This process, as demonstrated in this sentence, can avoid further cooling of sea surface, which in turn reduced sea surface heat loss to increase surface density, and thus stratifying the water column. This seems to exert rather positive feedback to stabilizing the cooling-induced AMOC slowdown. Please clarify. In general, positive/negative feedback discussed in this paragraph is hard to follow. Please clarify with more direct evidence/references.

*We agree with the reviewer that the discussion of feedbacks in this paragraph was unclear. Importantly, the old version suggested that sea ice cover is a negative feedback on SST changes, which is not correct. Sea ice cover prevents evaporation and heat loss to the atmosphere, stabilising the water column. We rewrote the paragraph focussed on stabilising mechanisms rather than feedbacks (lines 455-476).*

P13L11-13: As mentioned in previous comments, providing supportive evidence is of crucial importance since this is important positive feedback to the AMOC slow-down.

*We made more references to figures in the re-written paragraph.*

P13L15: please clarity and specify the positive and negative feedback mentioned here.

*We agree that this paragraph was unclear, and we rewrote it by using the clearer term 'stabilising process' (see above).*

P13L22-23: given the gradual decreasing radiative forcing, it is not clear whether it is the self oscillation or just an increased variability (small magnitude, 0.5Sv) as the system approaches the threshold. It appears that AMOC variance is of comparable or even larger magnitude during 6- 11kyr (Fig 5b). Is this also corresponding to self-oscillation?

*We are now more cautious with our statement and only write that variability is increased. The large variability at 6-11 kyr is related to density changes in the Irminger Sea. We discuss this in the new manuscript version in lines 386-393 in section 3.1, but we didn't see indications of oscillations:*

"After about 6 kyr, the changes in the North Atlantic density profile shifted the location of NADW formation: NADW formation moved south as vertical density profiles in the subpolar

east North Atlantic stabilised under a freshening of the surface and density profiles further south steepened due to surface cooling combined with subsurface warming (Fig. SI.7-9). These changes did not cause a step-change in AMOC strength, but freshwater and heat advection into the North Atlantic was reduced, sea ice expansion increased in the eastern North Atlantic, and AMOC variance (calculated over a moving 50-year window) was increased (Fig. 5)."

P13P26-33: as discussed, results from B.slow.b seem not to support the hysteresis behavior with respect to radiative forcing change. What about stability/sensitivity of the AMOC at ~6kyr in B.slow?

*We referred to hysteresis behaviour here because the radiative forcing that is required to cause the abrupt weakening of AMOC is not the same as the forcing required for strengthening it again. We are clearer in the revised manuscript. We did not test AMOC stability explicitly at 6 kyr because the high variability seems to cease once the density field has re-adjusted.*

P13L36: Orbital configuration consists of three orbital parameters. Their combinations in the chosen time slices are different but this does not mean the associated climatic impacts are significantly distinct, for instance, 21ka versus 0ka. It is thus better to show values of obliquity, precession, eccentricity and boreal summer insolation for the chosen time slices here, which would be helpful to clarify whether orbital forcing matters the transient behavior of the AMOC. A better approach to test roles of orbital configurations is to re-conduct such transient experiments based on orbital sensitivity, for example, high versus low obliquity experiments (e.g. experiments in Extended Data Table 1 of Zhang et al 2021).

*We now provide the orbital parameter values for each experiment in the SI. We were mostly interested here to see that changing the orbital configuration does not substantially alter the simulation results. It would be interesting to investigate the role of orbital changes for thermal thresholds in more detail in the future.*

P14L5-7: not a full list of key relevant papers. Please add Knorr & Lohmann 2007, Banderas et al 2012 and Zhang et al 2017. Re multiple stable AMOC states, the difference in the strength of the AMOC is significantly different with a magnitude of >5Sv. In this context, it appears that the metastable AMOC states proposed here are perhaps sub-states of the interglacial/glacial AMOC state. Given the low AMOC variability in Bern3D, I assume this might not be reproducible by full GCMs nor perhaps in proxies.

*We thank the reviewer for these additional relevant references, which we added to the paragraph. We also added further SI figures that show that each of the four persistent AMOC strengths is associated with different Greenland temperatures and North Atlantic sea ice extents, suggesting that they correspond to different climate states. Further, we agree, that it would be interesting to see this tested with a full GCM in future studies*

P14L12-16: what's the exact role of 'heat advection" in AMOC mode transition? A positive feedback, a trigger or else? It would be good to have a clearer description here to specify the importance of heat advection.

*Yes, we clarified this. Changes in heat convergence only seem to stabilise density profiles in the North Atlantic rather than causing AMOC shifts. We added these two paragraphs to section 3.1 (lines 386-397 and 453-457):*

"After about 6 kyr, the changes in the North Atlantic density profile shifted the location of NADW formation: NADW formation moved south as vertical density profiles in the subpolar east North Atlantic stabilised under a freshening of the surface and density profiles further south steepened due to surface cooling combined with subsurface warming (Fig. SI.7-9). These changes did not cause a step-change in AMOC strength, but freshwater and heat advection into the North Atlantic was reduced, sea ice expansion increased in the eastern North Atlantic, and AMOC variance (calculated over a moving 50-year window) was increased (Fig. 5). Transport of heat and salinity into the North Atlantic decreased (Fig. 5f, g), which reduced North Atlantic SST and SSS (Fig. 5e). The reduced influx of subtropical surface waters also caused abrupt cooling and freshening in the Irminger Sea (Fig. SI.8). At 24 kyr, the AMOC had weakened to ~14.5 Sv and sea ice cover extended south of the Irminger Sea (Fig SI.11)."

"The simulated step changes in AMOC strength in our simulations were thus the response to gradual surface cooling and freshening, and occurred when NADW formation shifted southwards. The resulting redistributions of heat and salinity caused sudden shifts in the vertical density profiles and sea ice expansion which consolidated the new circulation mode (Ando and Oka, 2021)."

P14L26-27: it is not true. For instance, Zhang et al 2017 applying a fully coupled AOGCM proposes that atmospheric CO2 levels are of control for glacial AMOC bi-stability.

*Our intention here was to understand why Oka et al. (2021), specifically, required a stronger forcing in the southern hemisphere for thermal thresholds to arise, while we see thermal thresholds in our model under a globally uniform forcing. We clarified this and now also refer to Zhang et al. (2017) (lines 534-595).*

P14L28-30: this may be true if comparing with other EMICs or simple models but not for GCM. Please clarify.

*This was a wrong conception, we removed this statement from the manuscript.*

P14L35: please provide modeling results or relevant literatures to support this statement especially regarding poleward moisture transport. It appears to me Fig 5e would be the right panel to refer to given the different trends between Atlantic and North Atlantic freshwater convergence. Sentences in P11L13-14 seem already touch this point, but it requires future clarification to link them to moisture transport and so on.

*The statement on moisture transport was specific to our model. The wind field is constant but the water holding capacity of air decreases with the temperature decline. Hence, less moisture is transported polewards by the large-scale atmospheric circulation. We clarified this in the text (lines 543-558). We also added more specific metrics of the changing water balance to the figures. We added SSS and SST timeseries for the Caribbean and Irminger Sea, as well as marine freshwater fluxes across latitudes 37.5°N and 62.5°N to Fig. 5, and spatial changes of P-E in the North Atlantic to the SI.*

P15: it is good to see the discussion about potential impacts of other parameters, especially ice sheet topography and associated wind, on the simulated AMOC change in different transient runs. In a glacial cycle, both changes in radiative forcing (e.g. CO2) and wind circulation/gateway caused by ice volume changes play a role in the strength/stability of the AMOC (Hu et al., 2011, Zhang et al 2014, 2017, 2021). Of most relevance here is their opposite impacts on the strength of the AMOC through glacial cycles in comparison to the thermal forcing (Barker and Knorr 2021). In this study, the authors investigated the roles of changes in radiative forcing in AMOC stability, which is the half story of AMOC multi-equilibria in glacial cycles. How do changes in those key parameters influence the results of A experiments? I would be happy to see more comprehensive discussion around this here as well as in Section 3.4 and 3.5. Perhaps, Section 3.3-3.5 can be integrated to one section to highlight and discuss the current understanding of AMOC stability, impacts of current model limitation on the current results and data-model comparison, and their implications and future perspectives.

*We followed the reviewer's advice and added more discussion of AMOC stability and model limitations by combining sections 3.3 and 3.5 (lines 509-664).*

P15 L14-15: Please add relevant reference to "different representations of processes affecting AABW density changes".

*We added a reference here (line 627):* "(e.g. brine rejection, Bouttes et al., 2011)"

P17 Figure 9: it would be good to flip y-axis of d18O curve upside down, given the tradition of plotting LR04/sea level curves.

*We inverted the y-axis of the $\delta^{18}O$ panel as suggested.*

References

[revised manuscript text omitted]

---

## Author Response (AR2)

*Dear Christo,*

*Thank you for the positive feedback. We revised the manuscript based on the reviewers' comments and provided point-by-point replies to each reviewer comment and amended the manuscript accordingly. Thanks to the reviewers' and your input, we believe this review process has greatly improved the manuscript. We hope the manuscript is now clear and comprehensive.*

*Best wishes,*
*The author team*

Reviewer #1

First of all, I apologise for the delayed report!

I thank the authors for the revision of the manuscript and the careful addressing of my comments. I find that most of my comments have been addressed satisfactorily. I have a few remaining points, mostly editorial and some necessary clarifications.

*We thank Dr. Marlene Klockmann for her positive assessment and constructive comments.*

l.64-68: I find this sentence a bit confusing. The first half (easier testing) is due to computational resources, right? The second half refers to the complexity of the model system which is being tested. And we have already found indications for threshold over a large range of complexities (as you correctly mention in the sentence before). It needs to be more clear what the sentence wants to say. Perhaps it is just the connection of the two halves with the "but" that makes me stumble.

*We divided the sentence into two and moved it to the end of the introduction, where it fits more clearly.*

l.87-88 please add Klockmann et al 2020 (https://doi.org/10.1029/2020GL090361) for completeness here

*Done.*

l.113-115: can your simulations be compared more directly to the proxies? And why exactly? In l.600-602 you say you do not expect a direct agreement with proxies.

*We expanded on these two points in the text (lines 110-113 and 613-616 in new manuscript version):*

*"While providing crucial process understanding, the limited simulation length makes direct comparisons of these simulations to proxy timeseries challenging, which is required to assess the role of these processes in glacial-interglacial AMOC changes."*

*"Since we chose to focus only on radiation driven AMOC changes in our experiments, while in reality AMOC was also influenced by freshwater flux changes, particularly during Heinrich events, we would not expect a close model-data match with reconstructed millennial-scale AMOC changes in the paleo-records."*

l.239: "lower three panels" instead of "lower two panels"

*Done.*

Fig.3/ l.295-299: I think the caption should refer to the modes in simulation A3 or simulation set A and not to the first 30kyrs of simulation B.slow? Also, please make the caption consistent with "Top:" and "Bottom:" as you did in the caption for Figure 2 (easier to read). Also, the modes in Fig.2 run from right to left, while in Fig.3 they run from left to right. Consider having them in the same order in both figures

*Done.*

l.538-541: Which simulation do you refer to, here? Also B.slow?

*This is a summary of the observed behaviour in all simulations, A and B. We mention this explicitly now.*

l.543-558: I find this whole paragraph difficult to read and follow. Is this meant in contrast to Oka et al 2021? Also some sentences don't work. Please rewrite for clarification!

*We now specify the models we discuss in each sentence. We also moved the last sentences of this paragraph into the next paragraph to improve clarity.*

l.600-602: Still, I find the comparison in Fig. 7a and 7b very impressive! Also, this sentence is somehow in contrast to l.113-115, where you seem to imply that your simulations can be compared more directly to proxies than other simulations (misunderstanding?)

*Yes, this was unclear. Our computationally-efficient model can produce long time series that are required for model-data comparisons for glacial-interglacial time scales but our forcing (specifically no freshwater hosing) prevents a model-data comparison of millennial-scale AMOC variability. Instead, our comparison focuses on long-term AMOC shifts during glacial cycles. We clarified this now in the introduction and discussion (lines 110-113 and 613-617 in the new manuscript version ):*

*"While providing crucial process understanding, the limited simulation length makes direct comparisons of these simulations to proxy timeseries challenging, which is required to assess the role of these processes in glacial-interglacial AMOC changes."*

*"Since we chose to focus only on radiation driven AMOC changes in our experiments, while in reality AMOC was also influenced by freshwater flux changes, particularly during Heinrich events, we would not expect a close model-data match with reconstructed millennial-scale AMOC changes in the paleo-records. Still, we can compare the long-term evolution of AMOC strength in our simulations and the reconstructions."*

Fig.9: Do the grey bars also correspond to MIS3 and MIS6? If yes, this could be added.

*Yes, we added this info to the caption now.*

Reviewer #2

This is the second round of my review on the manuscript "Multiple thermal AMOC thresholds in the intermediate complexity model Bern3D" by Adloff et al. I'm glad to see the revised version that answers most of my previous comments well, and has been largely improved in the demonstration. I believe this work provide new understanding of the AMOC thermal threshold during glacial cycles. Nevertheless, I still have some comments that shall be considered before my full support for its publication.

*We thank Dr Xu Zhang for the thorough review and the additional suggestions.*

Implications of abrupt AMOC change at ~27 kyr in B.Slow. In line 410-413, the authors argued that a weak bipolar seesaw (i.e. sea-ice retreat/warming in the Southern Ocean) could be identified during the biggest AMOC weakening, which is hard for me to confirm this from Fig. 5. I would rather suggest that the simulated cooling/sea ice expansion in North Atlantic and Southern Ocean at ~27 kyr are similar to glacial inceptions for instance MIS5-4 transition, when bipolar regions are characterized by significant cooling together with AMOC shoaling. This further indicates that AMOC-induced bipolar thermal seesaw might just be in a second order during glacial inceptions while decreasing radiative forcing (e.g. insolation, CO2, etc) is the dominant one, different from glacial conditions with mild insolation changes (e.g. MIS3).

*We agree that the bipolar seesaw effect is too weak to counteract the negative radiative forcing. We rephrased the section to avoid any confusion (lines 414-418 in the new manuscript version):*

*"The biggest AMOC weakening at ~27 kyr was also accompanied by a very weak bipolar seesaw effect (Stocker and Johnsen, 2003), which caused a temporary decline in sea ice coverage in the Atlantic sector of the Southern Ocean (Fig. 5). This sea ice decline, however, was too small to reduce the radiation-driven sea ice increase in the longer term."*

Line 80-81: Please categorize references here to specify to 1) stability/sensitivity of the AMOC and 2) AMOC self-oscillation. Please also cite Zhang et al. (10.1038/nature13592) for the former. This literature elaborates roles of wind circulation associated with ice sheet height changes in modulating the AMOC sensitivity/stability during glacial periods, and can also be referred to in the discussions for instance in line 554-558, 575-580, etc.

*Done.*

Line 87: Zhang et al., 2014 (10.1002/2014GL060321) is not a proper reference here since they did not resolve AMOC oscillations in their model. Instead, Zhang et al (10.1038/s41561-021-00846-6) should be cited in which simulated AMOC self-oscillations

are directly associated with successive DOs during MIS3. Please also update the citation in other relevant parts.

*Done.*

Line 90: please also cite Ganopolski & Rahmstorf (2001, Nature), which from my point of view is of direct support to this statement, but not for statement in Line 574-576.

*We added the reference to the introduction as suggested but also keep it in the discussion because they discuss differences of AMOC stability under LGM and PI conditions.*

Roles of earlier enhancement of AABW (associated with radiative forcing) in establishment of LGM ocean circulation. Fig 5 provides a good example to emphasize this point, which shall also be re-emphasized and discussed in Line 622-646, etc. There are dozens of literatures discussing such point but by snapshot experiments, for instance, Zhang et al. 2013 (10.5194/cp-9-1-2013), Galbraith & de Lavergne 2018 (10.1007/s00382-018-4157-8), which could be referred to in such discussion together with the transient modeling outputs in this study.

*We now mention this in the discussion of Fig. 5 and in the discussion of our model limitations (lines 419-422 and lines 594-597 in the new manuscript version) as suggested:*

*"The volume of AABW in the deep Atlantic influences AMOC stability (Zhang et al., 2013, Galbraith and Lavergne, 2019). Thus, the spread of AABW into the deep North Atlantic after the first AMOC shift at ~24 kyr might have preconditioned the AMOC for the following shift at ~27 kyr in B.slow."*

*"Northern Hemisphere ice sheets also affect the composition and volume of AABW through teleconnections (Galbraith and Lavergne, 2019), and the buoyancy difference between AABW and NADW, as well as their fraction in Atlantic deep water, have been found to precondition AMOC stability (Zhang et al., 2013)."*

Reviewer #3

The authors have made a huge effort in the revision and also managed to address all the comments I've wrote in the first round. I only have some minor comments left and happy to recommend the paper to be accepted for publication at the Climate of the Past.

Cheers,
Sam

*We thank Dr. Sam Sheriff-Tadano for the detailed and constructive comments on the manuscript.*

1. Relation of simulations A and B
Is simulation B.slow similar to A8? Sometimes I got confused when comparing results of B.slow and A2 or A3 since B.slow doesn't show a clear AMOC mode shift as in A2 and A3 in Fig. 1. Considering the speed of changes in the radiative forcing, I feel that B.slow and A8

are the closest. In that case, the fact the B.slow basically showing 2 or 3 modes makes sense. If the authors agree on this, it might be worth pointing it out in the Method section. One sentence would be sufficient. (Sorry if it was already explained somewhere..)

*This is a good point. We added a sentence to the Methods section as suggested:*

*"For comparison, the magnitude of this forcing is on the upper end of the range explored in simulation set A (A6-A8)."*

2. Role of Southern Ocean on the thermal threshold (L473-476 & L534-541)
The authors came to the conclusion that the North Atlantic processes are essential for the changes in the AMOC based on their analysis on salt transport, sea ice and deepwater formation regions. I agree to this in some sense, but also feel it's bit early to rule out the role of Southern Ocean (L473-476 & L534-541). To make this statement, I think additional experiments are required as in Oka et al. (2021). A good example is Oka et al. (2012) and (2021). In the 2012 study, they assumed that the thermal threshold was mostly related to the drastic shift of sea ice and the NADW formation region based on analysis. However, they found that this wasn't the case in their sensitivity experiments in the 2021 paper. While Fig. 5f doesn't show big changes in meridional salt flux across 32S, Fig. SI13 does show some salinity anomaly entering the Atlantic basin from Southern Ocean. Please amend these sentences (or some of the wordings) in a modest way so that the effect from the Southern Ocean cannot be ruled out at the moment.

*We agree. We weakened our statement and added two sentences (lines 548-551 in the new manuscript version) to clarify:*

*"Thus, in our model, Southern Hemisphere cooling does not need to exceed the cooling of the Northern Hemisphere to affect AMOC but further sensitivity tests would be required to establish the relevance of cooling in each hemisphere separately (as shown in Oka et al., 2021)."*

L299: Sorry, if it was explained in the response letter, but wasn't this figure created using results of the simulation A3?

*Yes, this was wrong. We corrected the caption.*

L383-384: Please explain the cause specifically. (e.g. due to fresher SSS and colder deep ocean temperature)

*Done.*

L386-389: Probably better to separate the sentence into two in my opinion. First one describing the shift of NADW formation region using Fig. S10 and S11 (is it correct?). Second one explaining the cause of it.

*We shortened the sentence and hope it is now easier to comprehend (lines 392-394 in the new manuscript version):*

*"After about 6 kyr, NADW formation moved south as surface freshening stabilised vertical density profiles in the subpolar east North Atlantic and density profiles further south steepened due to surface cooling combined with subsurface warming (Fig. SI.7-9)."*

L390 & L393; Are these sentences explaining similar thing? If so, please remove one of them to shorten the manuscript. Also shortening the manuscript is encouraged elsewhere.

*Done.*

L399: Probably better to say; a net increase in precipitation minus evaporation (P-E) led to …

*Done.*